# A peripheral epigenetic signature of immune system genes is linked to neocortical thickness and memory

Virginie Freytag[1,2], Tania Carrillo-Roa[3], Annette Milnik[1,2,4], Philipp G. Sämann[3], Vanja Vukojevic[1,2,5], David Coynel[2,6], Philippe Demougin[1,2,5], Tobias Egli[1,2], Leo Gschwind[2,6], Frank Jessen[7,8], Eva Loos[2,6], Wolfgang Maier[7,9], Steffi G. Riedel-Heller[10], Martin Scherer[11], Christian Vogler[1,2,4], Michael Wagner[7,9], Elisabeth B. Binder[3,12], Dominique J.-F. de Quervain[2,4,6,*] & Andreas Papassotiropoulos[1,2,4,5,*]

Increasing age is tightly linked to decreased thickness of the human neocortex. The biological mechanisms that mediate this effect are hitherto unknown. The DNA methylome, as part of the epigenome, contributes significantly to age-related phenotypic changes. Here, we identify an epigenetic signature that is associated with cortical thickness ($P = 3.86 \times 10^{-8}$) and memory performance in 533 healthy young adults. The epigenetic effect on cortical thickness was replicated in a sample comprising 596 participants with major depressive disorder and healthy controls. The epigenetic signature mediates partially the effect of age on cortical thickness ($P < 0.001$). A multilocus genetic score reflecting genetic variability of this signature is associated with memory performance ($P = 0.0003$) in 3,346 young and elderly healthy adults. The genomic location of the contributing methylation sites points to the involvement of specific immune system genes. The decomposition of blood methylome-wide patterns bears considerable potential for the study of brain-related traits.

[1] Division of Molecular Neuroscience, Department of Psychology, University of Basel, CH-4055 Basel, Switzerland. [2] Transfaculty Research Platform Molecular and Cognitive Neurosciences, University of Basel, CH-4055 Basel, Switzerland. [3] Department of Translational Research in Psychiatry, Max Planck Institute of Psychiatry, D-80804 Munich, Germany. [4] Psychiatric University Clinics, University of Basel, CH-4055 Basel, Switzerland. [5] Department Biozentrum, Life Sciences Training Facility, University of Basel, CH-4056 Basel, Switzerland. [6] Division of Cognitive Neuroscience, Department of Psychology, University of Basel, CH-4055 Basel, Switzerland. [7] German Center for Neurodegenerative Diseases (DZNE), D-53175 Bonn, Germany. [8] Department of Psychiatry, University of Cologne, Medical Faculty, D-50924 Cologne, Germany. [9] Department of Psychiatry, University of Bonn, D-53105 Bonn, Germany. [10] Institute of Social Medicine, Occupational Health and Public Health, University of Leipzig, D-04103 Leipzig, Germany. [11] Center for Psychosocial Medicine, Department of Primary Medical Care, University Medical Center Hamburg-Eppendorf, D-20246 Hamburg, Germany. [12] Department of Psychiatry and Behavioral Sciences, Emory University School of Medicine, Atlanta, Georgia 30322, USA. * These authors jointly supervised this work. Correspondence and requests for materials should be addressed to V.F. (email: virginie.freytag@unibas.ch) or to A.P. (email: andreas.papas@unibas.ch).

Human cortical thickness, a brain morphometric measure that is linked to cognitive functioning, reflects the amount of neurons and neuropil in the horizontal layers of the cortical columns that are responsible for the organization of cortical connectivity[1–3]. Recent data suggest a monotonic decrease in cortical thickness (cortical thinning) from preschool age throughout the lifespan[4], but previous studies have also described patterns of regional increase in cortical thickness during childhood[5–7].

Studies in twins and in unrelated individuals provide consistently high heritability estimates for cortical thickness (~80%), demonstrating the importance of naturally occurring genetic variation for this physiological trait[8,9]. Despite the well-known and substantial impact of age on cortical thinning, the biological mechanisms that mediate this effect are hitherto unknown. It is reasonable to assume that age-related, dynamic processes, such as epigenetic changes, represent good candidates for such mediators.

DNA methylation, the most extensively studied epigenetic modification to date, regulates important processes such as imprinting, chromosomal inactivation and gene expression[10]. Age represents one of the most potent factors known to correlate with physiological variation of global DNA methylation[11,12]. High-throughput quantification of DNA methylation at several hundreds of thousands of C-phosphate-G (CpG) sites has detected numerous CpG loci across various tissues undergoing differential methylation with age[13–16]. Interestingly, such loci have been identified within regulatory regions of genes that are known to undergo differential expression in such age-related conditions as Alzheimer's disease[13] and cancer[17]. Recently, DNA methylation markers predicting chronological age were shown to correlate with all-cause mortality[18]. DNA methylation levels can also be influenced by genetic variations[19,20] and age-related DNA methylation signatures represent heritable traits[18].

Thus, the existing data suggest that peripherally measured DNA methylation patterns might contribute to the identification of molecular underpinnings of age-related complex traits relevant to health and disease.

Here, we investigated the relation between peripherally measured DNA methylation and cortical thickness in healthy young adults. In a first step, we performed Independent Component Analysis (ICA)-based decomposition of whole-blood methylomic profiles to identify independent signatures of physiological variation of global DNA methylation. ICA is a decomposition method, which provides a representation of complex relationships arising from high-dimensional data, such as genome-wide expression[21,22] and brain imaging data[23]. After ICA-based decomposition, the identified methylation patterns were first tested for association with age. Age-associated methylation patterns were subsequently tested for correlation with global cortical thickness and, in case of such correlation, mediation analysis followed to assess whether these methylation patterns mediated significantly the effect of age on cortical thickness. Significant findings were subjected to further analyses, including functional annotation of CpGs contributing to the observed methylation patterns, testing for pattern association with region-specific cortical thickness and cognitive performance, and a genome-wide investigation of common genetic variations (single nucleotide polymorphisms, SNPs) that contribute to the variability of the methylomic patterns.

## Results

### ICA-based identification of methylomic patterns.
We performed methylomic profiling (Illumina 450K Human Methylation array) of blood samples collected from $N = 533$ healthy young individuals (Supplementary Table 1). After quality control, DNA methylation levels (DNAm) were quantified at 397,947 autosomal CpG sites and subsequently corrected for sex and sources of variation inferred from Surrogate Variable Analysis (see Methods).

Next, we performed ICA to achieve a low-dimensional representation of genome-wide methylation profiles. Following the ICA paradigm introduced first for gene expression data analysis[21], an individual's methylomic profile is treated as a mixture of latent variables (that is, methylomic signatures), each reflecting a combination of biological processes and exerting independent effects on DNAm. Specifically, ICA provides a representation of these signatures by decomposing the original DNAm signals into components, whose statistical inter-dependence is minimized. This property is typically achieved by favouring heavy-tailed non-gaussian distribution of the components' loadings; thus each component is characterized by a restricted set of CpGs exhibiting loadings at the extreme of the distribution. Simultaneously, each component is characterized by its representation across the study sample, giving rise to individual methylation patterns. Each of these patterns is a low-dimensional representation of a global mode of DNAm variations. Importantly, these patterns can be tested for association with traits of the study sample (Fig. 1a).

Using ICA decomposition, we obtained a total of $k = 126$ independent components (see Methods). The majority of the inferred components ($n = 111$) were driven by single individuals contributing to more than 10% of the pattern's variability. Given that such components represent rather singular modes of variation[24], subsequent analyses were restricted to the remaining 15 components. These components represent global modes of DNAm variation across the individuals of the study population.

### Methylomic patterns related to age and cortical thickness.
Participants from the methylomic profiling study underwent brain magnetic resonance imaging (MRI)(Supplementary Table 1). Global measures of cortical thickness—that is, the distance between the grey matter and white-matter boundary and the pial surface—were obtained using cortical surface-based analysis implemented in FreeSurfer (see Methods), for $N = 514$ participants. Consistent with previous findings in healthy young adults[4,25], cortical thickness was negatively correlated with age ($r = -0.27$, $P = 3.12 \times 10^{-10}$).

Two out of 15 ICA methylomic patterns (termed ICA1 and ICA2) were significantly correlated with age, after Bonferroni correction for 15 comparisons (ICA1: $r = 0.54$, $P_{nominal} = 1.54 \times 10^{-42}$, $P_{corrected} = 2.31 \times 10^{-41}$; ICA2: $r = 0.29$, $P_{nominal} = 4.68 \times 10^{-12}$, $P_{corrected} = 7.02 \times 10^{-11}$; Fig. 1c and Supplementary Table 2). These methylomic patterns were also significantly associated with cortical thickness (ICA2: $r = -0.24$, $P_{nominal} = 3.86 \times 10^{-8}$, $P_{corrected} = 5.79 \times 10^{-7}$; ICA1: $r = -0.14$, $P_{nominal} = 0.00162$, $P_{corrected} = 0.0243$; Fig. 1b and Supplementary Table 2). No significant correlation was observed between ICA1 and ICA2 ($r = 0.048$; nominal $P = 0.27$), suggesting that the corresponding independent components capture distinct methylomic processes.

To test whether the significant correlations between ICA2 and cortical thickness were merely attributable to the correlation between age and both types of measurements, age effects were partialled out from ICA2 and cortical thickness (see Methods). After this adjustment, a significant correlation was exclusively detected between ICA2 and cortical thickness ($r = -0.18$; $P = 6.55 \times 10^{-5}$, Supplementary Table 2). The correlation remained significant ($r = -0.17$; $P = 8.74 \times 10^{-5}$) also after correcting for individual white blood cell count (see Methods). We also examined which other available variables (that is, body

mass index, smoking, alcohol consumption, frequency of cannabis use) were significantly associated with *ICA2* in addition to age. Smoking frequency was also significantly associated with *ICA2* ($r = 0.17$, $P = 0.0001$) but not with cortical thickness ($r = -0.072$, $P = 0.11$). After adjusting *ICA2* for both age and smoking frequency, its association with cortical thickness remained nearly unchanged ($r = -0.17$). No significant correlations were detected between *ICA2* and alcohol consumption ($P = 0.97$), cannabis use ($P = 0.1$) or body mass index ($P = 0.25$).

In order to capture possible non-linear age effects, we also performed an *F*-test analysis to compare the fit of a model predicting cortical thickness from a fifth degree polynomial of age ($age + age^2 + \ldots + age^5$) to the fit of the same model augmented by *ICA2*. We observed a highly significant increase in adjusted $R^2$ with the addition of *ICA2* to the model ($F(1,507) = 15.6$, $P = 8.8 \times 10^{-5}$). Thus, the association between *ICA2* and cortical thickness is not driven by non-linear age effects.

We also used *in silico* annotation of blood cell types as described by Jaffe and Irizarry[26]. After this adjustment, *ICA2* associations with both chronological age and cortical thickness remained highly significant ($P = 2 \times 10^{-11}$ and $P = 8.3 \times 10^{-7}$,

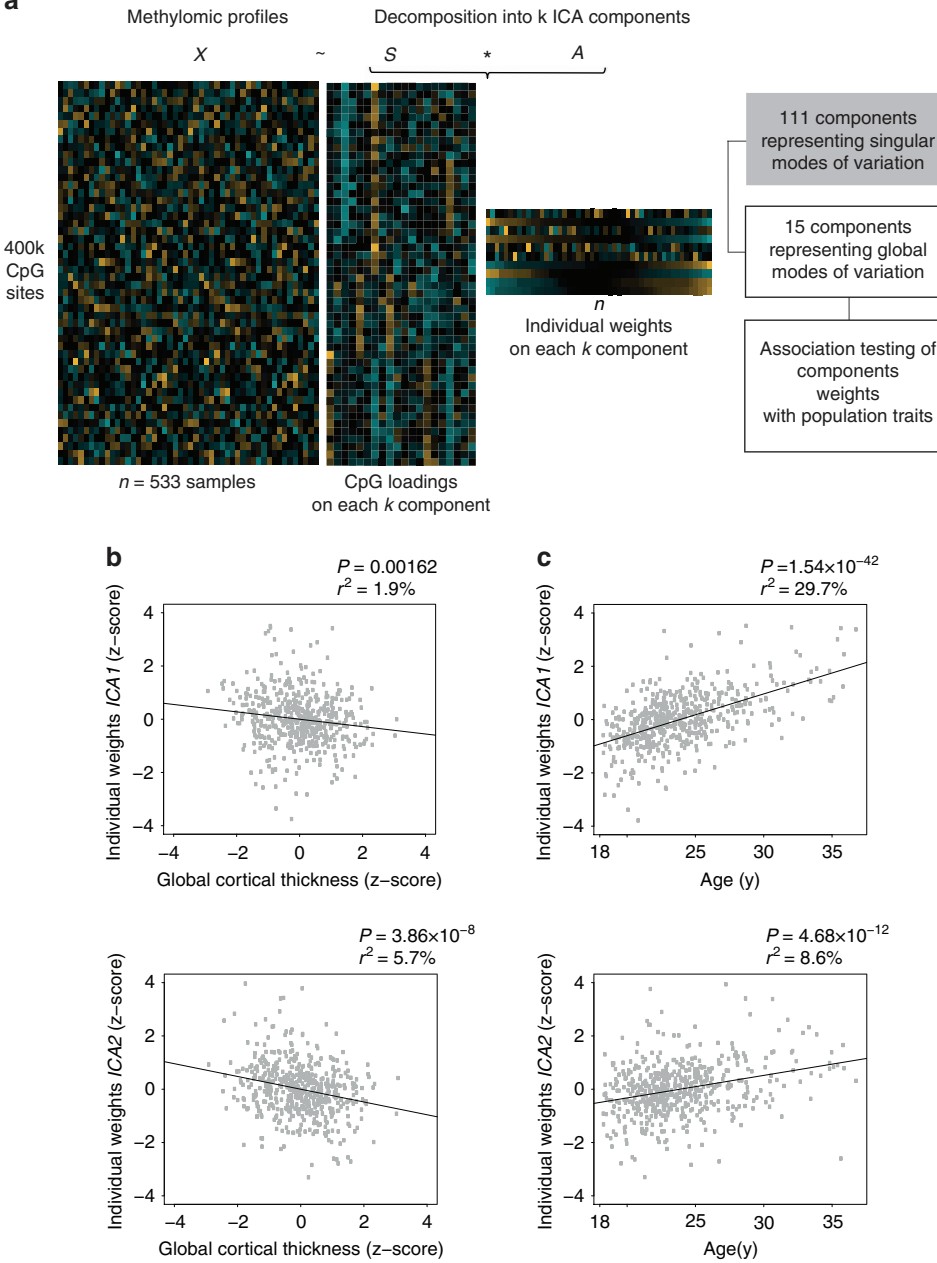

**Figure 1 | ICA-based identification of DNAm patterns.** (**a**) Schematic representation of the analysis workflow; ICA decomposition of genome-wide methylomic profiles (matrix *X*, $n = 533$ samples $\times$ 397,947 CpGs sites) into *k* independent components, simultaneously represented across CpGs (matrix *S* of CpGs loadings) and samples (matrix *A* of individual weights). A total of 15 components, whose corresponding weights represent global modes of DNAm across samples, were tested for association with cortical thickness and chronological age. (**b**) Two components, *ICA1* and *ICA2*, are significantly associated with cortical thickness. Horizontal axis: cortical thickness adjusted for sex, intra-cranial volume and MR-batches. Vertical axis: individual weights on *ICA* component. (**c**) *ICA1* and *ICA2* show significant association with chronological age. *P*: *P* value of association (Pearson's correlation, two-sided test); $r^2$: fraction of variance in component weights explained by chronological age (in %).

respectively). We also examined the association between *ICA1* and *ICA2* and chronological age in two publicly available data sets of purified blood cells ($N = 1,202$ monocyte samples, age range: 44–83, mean age: 60; $N = 214$ CD4+ T-cell samples, age range: 45–79, mean age: 59)[15]. In each data set, *ICA1* and *ICA2* were estimated as the linear combinations between *ICA1* and *ICA2* loadings, respectively (as inferred from the Swiss DNAm sample), and blood samples' DNAm values, adjusted for main confounders (see Supplementary Methods). In both cell-specific data sets, we observed a significant positive correlation between *ICA* patterns and chronological age (monocyte samples, $N = 1,202$: *ICA1*: $r = 0.67$, $P < 2.2 \times 10^{-16}$; *ICA2* $r = 0.32$, $P < 2.2 \times 10^{-16}$; CD4+ T-cell samples, $N = 214$: *ICA1*: $r = 0.70$, $P < 2.2 \times 10^{-16}$; *ICA2*: $r = 0.49$; $P = 8.6 \times 10^{-15}$), suggesting that the *ICA*–age correlations identified in whole-blood are also detectable in individual cell types. Altogether these results substantiate the lack of influence of blood cell counts on the reported associations. The correlation between cortical thickness and *ICA1*, that showed the strongest correlation with age, was not significant after adjusting for chronological age ($r = 0.01$, $P = 0.83$, Supplementary Table 2).

In addition to chronological age, we also calculated epigenetic cross-tissue- and whole-blood-based predictors in our sample as described by Horvath[27] and Hannum *et al.*[14], respectively. Both estimators yielded DNA methylation age values (that is, predictors for chronological age based on CpG methylation) that significantly correlated with actual participants' age (Horvath's predictor: $r = 0.70$, $P < 10^{-60}$; Hannum's predictor: $r = 0.71$, $P < 10^{-60}$). Neither predictor was associated with cortical thickness after adjustment for chronological age (Horvath's: $r = 0.04$, $P = 0.32$; Hannum's: $r = 0.01$, $P = 0.77$), suggesting that these predictors (like *ICA1* but, importantly, unlike *ICA2*) do not mediate the effect of age on cortical thickness.

Finally, we examined the association of *ICA2* with age and age-adjusted cortical thickness after covarying for 111 individuals who contributed more than 10% to 111 inferred components not further studied herein. Both associations remained highly significant (age: $P = 4.91 \times 10^{-12}$; age-adjusted cortical thickness: $P = 4.8 \times 10^{-5}$).

**Replication study**. To test the generalizability of the association between *ICA2* and cortical thickness, we studied an independent sample (termed herein the Munich sample) comprising 596 participants with major depressive disorder (MDD) and healthy controls (see Methods). The *ICA2* pattern was estimated as the linear combination between *ICA2* loadings (as inferred from the Swiss DNAm sample) and individual DNAm values of the Munich sample. In this independent sample, we observed a significant positive correlation between *ICA2* and chronological age ($N = 596$, $r = 0.48$, $P < 10^{-10}$) and a negative correlation with global cortical thickness ($N = 596$, $r = -0.31$, $P < 10^{-10}$). After adjustment for chronological age and controlling for potential confounders (diagnosis, sex, intracranial volume, MRI batch effects, time difference between MRI examination and blood drawing), the association between *ICA2* and cortical thickness remained significant ($r = -0.094$, $P = 0.011$). The same analysis in a sub-sample of $N = 163$ participants younger than 40 years (that is, within an age range similar to that of the Swiss participants) revealed an almost identical effect size ($r = -0.19$, $P = 0.009$) compared to that observed in the Swiss sample.

***ICA2* partially mediates the age–cortical thickness relation**. *ICA2* showed significant positive correlation with age and negative correlation with global cortical thickness. To investigate

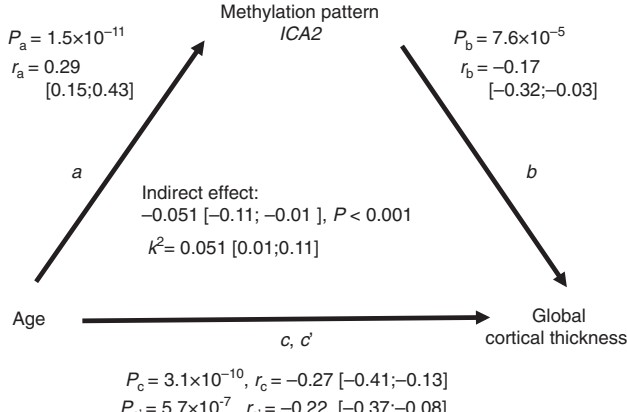

**Figure 2 | Mediation analysis of methylomic pattern *ICA2* on the association between chronological age and global cortical thickness.** Path *a* represents the effect of chronological age on *ICA2*. Path *b* represents the effect of *ICA2* on global cortical thickness after removing the effect of chronological age. Path *c* denotes the total effect of chronological age on global cortical thickness. Path *c'* represents the direct effect of chronological age on cortical thickness while controlling for the indirect effect (*a* multiplied by *b*). *r*: correlation coefficient; 99.9% confidence interval for the parameters are shown in brackets; *P*: *P* value of association. $k^2$: kappa-squared standardized maximum possible mediation effect.

whether *ICA2* mediates the negative correlation between age and global cortical thickness, we conducted a mediation analysis[28]. The association between chronological age and global cortical thickness was partially (that is, $k^2 = 5.1\%$ of the maximum possible mediation effect) and significantly mediated by the methylomic pattern *ICA2* (indirect effect $= -0.051$, $P < 0.001$) (Fig. 2).

***ICA2* is related to a specific pattern of cortical thickness**. Having detected an association between *ICA2* and global cortical thickness we next explored possible links between this methylomic pattern and regional variations in cortical thickness. Inter-individual variations in delineated brain regions often coincide with latent structural covariance patterns[29]. Exploratory factor analysis (EFA) allows depicting such distinct patterns of volumetric covariance among brain regions that can be subsequently tested for association with additional phenotypes of the population under study[30]. We therefore performed EFA, considering 68 regional brain measures of thickness (34 per hemisphere) obtained from automated parcellation of the cerebral cortex (Desikan-Killiany atlas)[31–33]. Before analysis, effects of intra-cranial volume, sex, processing batches and age, which possibly drive global correlations among brain regions, were regressed out from individual measures (see Methods). Using parallel analysis[34], we determined eight extractable factors, altogether accounting for 48.9% of variance across regional measures (Supplementary Data 1, see Methods). Factor extraction was followed by varimax orthogonal rotation. Subjects' factor scores were subsequently tested for association with the age-adjusted *ICA2* pattern. After correction for multiple testing, we identified one factor score, *F6*, that showed significant correlation with *ICA2* ($r = -0.13$, $P = 0.00314$, Bonferroni-adjusted $P = 0.025$ for eight tests conducted)(Fig. 3a and Supplementary Table 3). This factor, accounting for 4% of variance in cortical thickness measures, was characterized by a spatial pattern comprising mainly temporal areas (loadings > 0.3), with the highest loadings observed for left and right temporal poles and

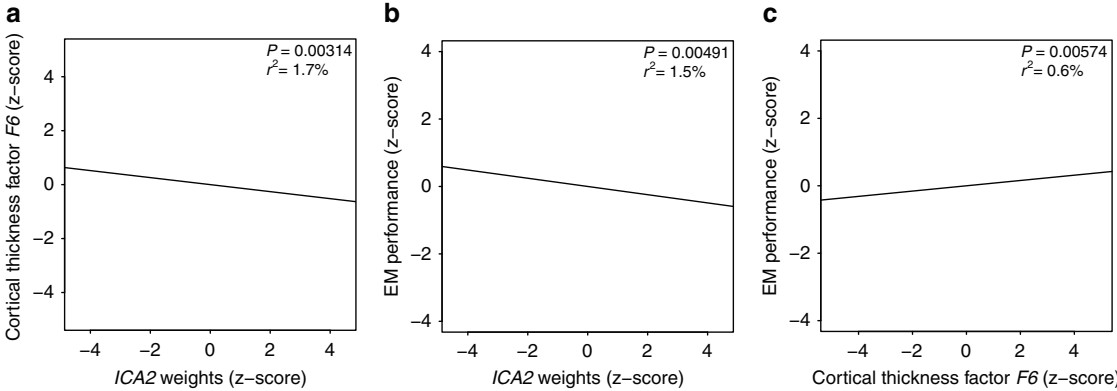

**Figure 3 | Correlations between *ICA2* weights, EM performance and cortical thickness score *F6*.** (**a**) Correlation between cortical thickness factor score *F6* and *ICA2* weights in the methylomic profiling sample. (**b**) Correlation between *ICA2* weights and EM performance in the methylomic profiling sample. (**c**) Correlation between cortical thickness factor score *F6* and EM performance in the combined sample ($N = 1,234$). Subjects from the methylomic profiling sample are shown in blue. *ICA2* and the EM/imaging phenotypes are adjusted for chronological age effects.

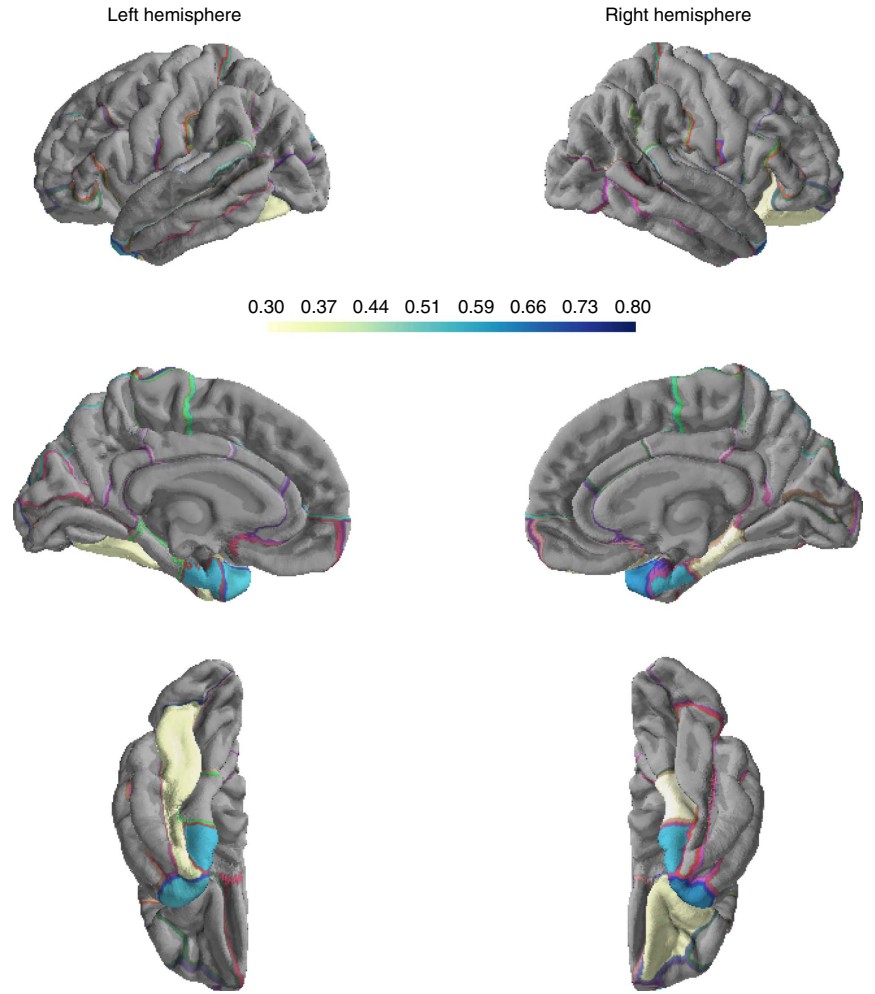

**Figure 4 | Regional cortical thickness loadings on factor *F6* associated with *ICA2* methylomic profile.** Absolute values for loadings are considered. Loadings $< |0.3|$ are not shown.

entorhinal cortices (mean loadings across the four temporal regions: 0.58) (Fig. 4 and Supplementary Data 1). We also run EFA under most conservative adjustment of the 68 regional brain measures of thickness for mean global thickness, to study whether the significant regional effects observed herein are fully explained by mean global cortical thickness. We observed high mean correlation between factor loadings across the two EFA solutions ($r = 0.78$); importantly, *F6* remained stable across the two solutions with an $r = 0.89$ ($P = 6.9 \times 10^{-24}$) between loadings before/after adjustment for mean thickness. In addition, *F6* scores

obtained from the mean thickness-adjusted EFA solution were still significantly associated with *ICA2* ($r = -0.09$, $P = 0.042$), suggesting that the results presented herein were not driven solely by global mean thickness.

Thus, higher values of *ICA2* are related to thinning of a circumscribed cortical pattern that harbours neuroanatomical correlates of episodic memory (EM). Therefore, we investigated the relationship between this methylomic pattern and EM. Behavioural assessment was obtained for a total of $N = 531$ subjects from the methylomic profiling study (see Methods, Supplementary Table 1). We detected a significant negative correlation between *ICA2* and EM performance ($r = -0.138$, $P = 0.00147$) (Supplementary Table 4). This association remained significant after partialling out age effects ($r = -0.122$, $P = 0.00491$) (Supplementary Table 4 and Fig. 3b) and after adjustment for white blood cell types abundance ($r = -0.105$, $P = 0.016$ after adjusting for age).

Complete MRI and EM assessments were obtained for $N = 512$ participants from the methylomic profiling sample (Supplementary Table 1). In this sample, the correlation between *F6* regional thickness score and EM performance was not significant ($r = 0.048$, $P = 0.283$)(Supplementary Table 5). In an additional independent sample of $N = 722$ healthy young adults (Basel imaging sample, Supplementary Table 1), who underwent identical MRI (in the same scanner) and cognitive assessment as the methylomic profiling sample (see Methods), the correlation between *F6* regional thickness score and EM performance was significant ($r = 0.102$, $P = 0.00617$)(Supplementary Table 5). Importantly, subjects' *F6* factor scores from this additional MRI sample were predicted based on the factor solution inferred from the methylomic profiling sample. In the combined sample ($N = 1,234$ participants), the correlation between *F6* and EM performance was significant ($r = 0.079$, $P = 0.00574$) (Fig. 3c and Supplementary Table 5).

**Functional and genomic characterization of *ICA2* CpGs.** Contributions to a given ICA component are commonly identified by selecting features (here: CpG sites) whose loadings, in absolute value, exceed a cut-off threshold of $n_\sigma$ standard deviations from the mean of the loadings' distribution[22]. In gene expression studies, such a typical threshold ranges between 2 and 3 (ref. 35). Given the pronounced multidimensionality of the methylomic profiles, we used a stringent cut-off of $n_\sigma = 4$, which led to the selection of 970 CpGs for *ICA2* (Supplementary Data 2). The selected *ICA2* CpGs mapped to 593 genes (see Methods). Among the 970 CpGs constituting *ICA2*, one marker (cg18055007) was part of the 353 Horvath age-predicting markers[27], and four (cg20822990, cg16054275, cg16867657, cg21139312) were part of the 71 CpGs included in Hannum's DNAm age model[14]. We also examined whether, and to what extent, *ICA2* CpGs ($N = 970$) overlapped with those reported as being differentially methylated ($N = 2,037$) in smokers[36–39]. This was the case for a small fraction (3%) of the *ICA2* CpGs.

Enrichment analysis for gene ontology (GO) terms, molecular pathways and gene expression patterns, as catalogued in the Molecular Signatures Database (MsiGDB, www.broadinstitute.org/gsea/msigdb/index.jsp), was performed using the GOseq algorithm[40], which corrects for multiple CpG mapping per gene (see Methods). Using an FDR threshold $< 0.05$ (Benjamini–Hochberg adjustment), analysis of *ICA2* revealed significant enrichment for 76 highly overlapping gene sets, which mainly encompassed genes related to immune system function, inflammatory response and hematopoietic system (Supplementary Data 3). To explore further the nature of the immune component related to cortical thickness, we compared

the DNA methylation of the 970 most prominent *ICA2* CpGs to that of blood cell subtypes and their progenitors using public data sets on 19 cell types (see Methods)[41]. We observed consistently highly significant correlations ($N = 970$ CpG sites, $P < 10^{-60}$ for all correlations) between average whole-blood DNAm values of the *ICA2* CpGs and all various cell subtypes examined (Supplementary Figs 3–5). The lowest correlation coefficients (albeit still highly significant with $P < 10^{-60}$) were observed for regulator and memory CD4 + T-cells (Supplementary Fig. 5). Generally, the correlation coefficients might suggest high concordance of the cortical thickness-related blood DNAm patterns with DNAm of B lymphocytes and of the common myeloid progenitor lineage, and relatively less concordance with DNAm of natural killer cells and T lymphocytes.

We next characterized the *ICA2*-contributing CpGs with respect to their topographical distribution across the genome (that is, island, shore, shelf, open sea regions) and to their relative location to gene transcripts. Given that a CpG site may map to multiple transcripts, each site was uniquely characterized according to following rules[42]: CpGs annotated within 1,500 bp upstream the transcription start site of at least one transcript were flagged as 'TSS'; CpGs not flagged as 'TSS' but located within a transcript (including 3'UTR, 5'UTR) were flagged as 'Genic'; all remaining CpGs were flagged as 'Intergenic'. We observed a significant shift in the distribution of CpG topographical categories as compared to the genome-wide background expectations ($\chi^2$ test $P = 2.7 \times 10^{-53}$) with 50% of all CpGs annotated as Open Sea, while Islands CpGs were clearly under-represented (10%)(Supplementary Fig. 1A). The distribution of CpG sites across genomic context categories differed from the genome-wide background distribution ($\chi^2$ test $P = 1.65 \times 10^{-6}$), with an increased fraction of 'Genic' CpGs and a decreased fraction of 'TSS' CpGs. We also observed a lower fraction of intergenic CpGs as compared to the background distribution (Supplementary Fig. 1B).

Finally, to test between-sample comparability of the identified ICA patterns, we performed ICA of the study population reported in Hannum *et al.*[14], which consists of 656 blood DNAm profiles of participants spanning a wide age-range (19–101 years, mean age: 64 years). We identified five ICA patterns that were significantly associated with age ($N = 656$, $P = 0.000043$ – $P < 10^{-60}$). We then examined the overlap between *ICA1* and *ICA2* CpGs identified in our sample and CpGs contributing to each of the Hannum age-associated IC pattern. A significant overlap with *ICA1* was observed for one pattern (termed here HICa, OR = 91, $P < 10^{-60}$). For *ICA2*, we observed a significant overlap with three age-associated Hannum patterns ($P = 1.9 \times 10^{-6}$ – $P < 10^{-60}$), with said overlap being particularly strong for one pattern (termed here HICb, OR = 49, $P < 10^{-60}$). The correlation of loadings between CpGs contributing to *ICA2* and CpGs contributing to the HICb pattern was positive and of substantial magnitude ($r = 0.87$, $P < 10^{-60}$). Thus, we observed highly significant between-sample overlap of ICA patterns despite the differences in age structure of the two populations.

**ICA2-derived multigenic score associated with EM performance.** Given that DNA methylation patterns per se represent complex traits[18,43], we studied the genetic underpinnings of the *ICA2* pattern. As for any genetically complex trait, several genetic variants are likely to contribute jointly to inter-individual variability of DNAm variation as represented by *ICA2*. Therefore, we employed gene set enrichment analysis (GSEA)[44–46] to disentangle biologically meaningful subsets of genetic contributions to *ICA2*.

**Table 1 | GSEA results for *ICA2* pattern.**

| Database | Gene set | No. of genes* | Nominal GSEA $P^{\dagger}$ | FDR |
|---|---|---|---|---|
| Gene ontology | Leukocyte differentiation | 38 | $7.1 \times 10^{-5}$ | 0.0124 |
| Biocarta | PYK2 pathway | 27 | $6 \times 10^{-4}$ | 0.0183 |
| Gene ontology | Lymphocyte differentiation | 26 | $8.6 \times 10^{-5}$ | 0.0234 |
| Biocarta | Keratinocyte pathway | 44 | $2 \times 10^{-4}$ | 0.024 |
| Gene ontology | Haemopoiesis | 71 | $3.22 \times 10^{-4}$ | 0.0407 |
| Gene ontology | Haemopoietic or lymphoid organ development | 73 | $2 \times 10^{-4}$ | 0.044 |

*Number of genes in gene set mapped by at least one SNP.
†Empirical enrichment $P$ value at a 75th percentile cut-off.

DNA from all individuals participating in the methylomic profiling study was processed on the Affymetrix Genome-wide Human SNP Array 6.0. After standard QC, correction for minor allele frequency and deviation from Hardy–Weinberg equilibrium, a total of 733,370 autosomal SNPs were used for association analysis (see Methods).

Age-adjusted single-marker $P$ values for association with *ICA2*, under an additive model, were tested for gene set enrichment using MAGENTA[44] (see Methods). Across the 1,411 tested sets we detected a significant over-representation of association signals (FDR < 0.05) in six gene sets mainly related to immune system regulation (Table 1). Given the substantial overlap between the identified gene sets, we further combined these sets into two gene groups with minimum overlap: genes from categories GO: Lymphocyte differentiation, GO: Leukocyte differentiation and GO: Haemopoiesis were grouped into GO:0048534 (Haemopoietic or lymphoid organ development) which comprised 73 unique genes; the two remaining gene sets, that contained 12 overlapping genes, Biocarta: Pyk2 pathway and Biocarta: Keratinocyte pathway were grouped into 'Pyk2/Keratinocyte pathway', which comprised 60 unique genes. These two distinct gene groups had one gene in common.

For each of the gene groups we calculated multilocus genetic scores to capture their contributions to individual *ICA2* variability. The scores comprised 39 and 33 significant SNPs mapping to an equal number of genes from the 'GO:0048534' and 'Pyk2/Keratinocyte' groups respectively (Supplementary Data 4 and 5). Genetic scores were weighted by the direction of effect of single-marker association statistics, resulting in positive correlation of each score with the *ICA2* pattern (see Methods). As expected, both scores correlated significantly with *ICA2* variability ('GO:0048534': $r = 0.53$, $P = 3.26 \times 10^{-40}$; 'Pyk2/Keratinocyte': $r = 0.45$, $P = 5.9 \times 10^{-28}$).

The genetic score derived from the Haemopoietic and Lymphoid Organ development set (GO:0048534) was significantly correlated with EM performance ($r = -0.10$, $P = 0.01$). No significant correlation was detected for the 'Pyk2/Keratinocyte'-derived genetic score ($r = 0.02$, $P = 0.7$).

To test the robustness of this association, we studied the correlation between the GO:0048534-derived genetic score and EM in four additional independent samples ($N = 3,346$): three samples, including subjects from the Basel imaging sample, comprised a total of $N = 2,603$ healthy young subjects who performed either a picture free recall or a word free recall task; an additional sample included $N = 743$ elderly healthy individuals who performed a word free recall task (age range: 74–91 years, see Methods and Table 2). The genetic score correlated negatively with EM performance, resulting in a significant combined association $P = 0.0003$ (Stouffer Meta-analysis, Table 2).

To test whether GSEA-derived genetic score SNPs are enriched for mQTLs of the *ICA2* CpGs, we first examined the location of

**Table 2 | Association of Haemopoetic or Lymphoid Organ development genetic score and EM-related traits in independent samples.**

| Sample | *N* | Age range | EM task | *r* | *P* |
|---|---|---|---|---|---|
| Basel cognitive | 1,445 | 18–35 | Pictures | − 0.062 | 0.00912 |
| Basel imaging | 534 | 18–35 | Pictures | − 0.02 | 0.32 |
| Zurich | 624 | 18–45 | Words | − 0.073 | 0.0349 |
| AgeCode | 743 | 74–91 | Words | − 0.076 | 0.0191 |
| Stouffer's method meta-analysis | | | | − 0.06* | 0.0003 |

*r*: Pearson's correlation coefficient. *P*: one-sided correlation test *P* value.
*sample size weighted *r*.

these SNPs relative to the 970 CpGs constituting *ICA2*. We observed a significant over-representation (53%, $P < 0.0002$) of gene score SNPs *in cis* (that is, $\pm 1$ Mbp) to *ICA2* CpGs as compared to a genome-wide random distribution (see Methods). Next, we performed mQTL analysis for each of the score SNP–*ICA2* CpG pairs. We observed significant deviation from the null uniform distribution with particular over-representation of genetic associations with effect sizes ranging from small to moderate (Supplementary Fig. 2). Thus, GSEA-derived SNPs collectively exert multiple genetic effects of small to moderate magnitude on the CpGs contributing to *ICA2*.

We also studied the association between the genetic score and cortical thickness in the methylomic sample ($N = 514$). No significant correlation was observed with cortical thickness ($r = -0.06$, $P = 0.08$).

## Discussion

In the present study we applied ICA decomposition of whole-blood genome-wide methylomic profiles in healthy young adults ($22.9 \pm 3.3$ years, mean ± s.d.) and detected a specific pattern of DNAm (*ICA2*) that was associated with cortical thinning and decreased EM performance. We also observed that a significant part of the well-known negative correlation between age and cortical thickness was partially mediated by *ICA2*. CpG sites that contributed to this methylation pattern mapped to genes involved in immune system regulation and inflammatory response.

Notwithstanding the robust and replicated findings presented herein, we would like to stress some limitations, which are inherent to the study design. First, the mediation analysis suggests that *ICA2* significantly, albeit partially, mediates the effect of age on cortical thickness. Given the associative nature of the data, we cannot exclude the possibility that the correlation observed between *ICA2* and cortical thickness might also be partially driven by additional non-modelled variables. Second, decomposition of genome-wide methylomic profiles comes at the

cost of specificity of the inferred solution towards the genomic localization of CpG markers. The detection of CpGs contributing to the methylomic signature relies on a fixed threshold on the distribution of the components' loadings. In our case, this approach allowed relating ICA2 broadly to genes involved in immune system function. However, the specific relationships between the identified marker sets and the phenotypes of interest can be studied only in downstream experiments focusing on single CpG sites. Third, the ICA model relies on the assumption that methylomic signals arise from a fixed set of independent sources. In the absence of a priori knowledge about the source signal, the number of inferred components must be determined empirically, which might impact negatively on generalizability. Integration of multiple-layers of molecular traits, such as genotypic data used in this study, is therefore important to address whether the identified patterns represent relevant features of the data set.

The cellular mechanisms underlying changes in cortical thickness are not entirely clear; however, they are most likely life phase-dependent. During development, cortical thinning might be related to events mirroring cortical maturation, such as synaptic pruning[47], whereas shrinkage of neurons, reductions of synaptic spines and lower numbers of synapses probably account for adult age-related cortical thinning[48]. In addition, myelination of lower cortical layers might cause the cortical mantle to appear thinner on MR scans[49]. This phenomenon might account for a substantial part of the observed cortical thinning during development. The correlation between cortical thickness and cognitive function also seems to be age-dependent. In adulthood and old age, cortical thinning is associated with a decline in cognitive function[3], whereas during development this relationship is dynamic with predominantly negative correlation between cognitive function and cortical thickness in early childhood to a positive correlation in late childhood and beyond[6]. Importantly, a substantial proportion of the strength of the relation between cortical thinning and cognitive decline in adults is attributable to the influence of age in each type of measure[3].

The association of ICA2 with cortical thickness and EM performance reported herein supports observations relating the peripheral immune system to brain morphology and cognition[50,51] and is coherent with the notion that the brain and its functions is directly linked to peripheral tissues relevant to the function of the immune system[52]. The data presented herein might suggest high concordance of the cortical thickness-related blood DNAm patterns with DNAm of B lymphocytes and of the common myeloid progenitor lineage, and relatively less concordance with DNAm of natural killer cells and T lymphocytes. Nevertheless, it is important to stress that we cannot draw any mechanistic conclusions about the relationship between peripheral methylation on the one side and cortical thickness and EM performance on the other, and that no further inference can be drawn towards the contribution of a specific immune cell type to the reported associations. Indeed, the mechanisms through which the peripheral immune system exerts an influence on the central nervous system remain elusive. Direct cytokine-induced central responses or indirect cytokine-mediated changes within the central nervous system via activation of vagal-nerve afferents are being discussed among possible scenarios[53,54]. Of note, methylation sites related to IL6R (encoding interleukin 6 receptor), ZC3H12D (encoding zinc finger CCCH-type containing 12D) and CD4 (encoding CD4 molecule) are listed among the top ten ICA2 contributing CpGs (Supplementary Data 2) in our data. The products of these genes are centrally implicated in cytokine signalling, mRNA stability of cytokine genes and immunological response. It will be interesting

to investigate whether direct measurement of the immune factors implicated herein along with traditional blood markers of the immune system will provide additional information with regard to the relation between these immune factors and cortical thickness. We speculate that this might not be the case, given the substantial volatility of such direct measurements, which mostly reflect acute state of the immune system, whereas methylation profiles reflect, at least partially, a record of past immune regulation. Nevertheless, further experimental work is warranted to test this hypothesis.

Inter-individual variability in blood cell composition is known to influence whole-blood DNAm measurements[55]. In our population of healthy young adults, no significant association between blood cell sub-types and cortical thickness or EM performance was observed (Supplementary Table 6). Moreover, the associations between ICA2, cortical thickness and EM were significant also after correction for blood cell composition. In addition, we observed a significant positive correlation between ICA patterns (ICA1 and ICA2) and chronological age in the examined blood cell-specific data sets. Thus, it is unlikely that the detected associations are driven by inter-individual variability in composition of blood cell types.

In addition to studying methylation patterns, we also performed a genome-wide SNP-based analysis of ICA2. The reasons for this analysis were two-fold: (1) Given the fact that genetic variation is related to DNA methylation, we tested whether ICA2-related genetic variation can be used as a proxy for DNA methylation in larger samples, where such epigenetic measures were unavailable. (2) We hypothesized that the biological processes revealed through gene set enrichment would be similar regardless of the nature of the data input (that is, genetic versus epigenetic variation). Interestingly, the SNP-based analysis of ICA2 revealed a robust association between variants of genes involved in the regulation of the immune system and EM in independent cohorts of young and elderly healthy adults. This suggests that the association between ICA2, which reflects epigenetic variation, and EM performance is, at least partially, genetically driven.

In conclusion, we adopted an ICA approach to achieve a tractable and biologically meaningful representation of genome-wide methylation profiles that are amenable to association testing. To this end we searched for methylomic profiles that arise from putatively independent biological processes, each reflected by a restricted number of CpG sites. By decomposing genome-wide DNAm profiles we identified an epigenetic mark of immune system genes linked to cortical thickness and to human memory. The well-known effect of age on cortical thinning is partially mediated by this epigenetic mark, and its genetic underpinnings also point to genes involved in immune system regulation. Thus, the decomposition of blood methylome-wide patterns bears considerable potential for the study of brain-related physiological traits. For example, peripheral markers of systemic inflammation are associated with reduced grey matter volume, both in midlife adults[50] and in the elderly[56]. Moreover, such grey matter reduction seems to mediate the negative effects of peripheral inflammation on age-related cognitive decline[50]. It will be interesting to investigate whether the peripheral DNAm profiles identified herein might be used to differentiate between physiological and pathological age-related cognitive decline and cortical thinning.

## Methods

**Samples.** *Methylomic profiling sample.* This sample is part of an ongoing, continuously recruiting imaging genetics study of healthy young adults in the city of Basel, Switzerland. Aim of the study is to recruit large samples of healthy young adults for assessing cognitive performance measurements, personality traits,

functional and anatomical MRI and genetics (based on saliva DNA) at the time-point of the main investigation. Advertising for the main investigation was done mainly in the University of Basel. Subjects were re-invited via email or at the time-point of the main investigation to an additional blood and saliva sampling. The time point of this second investigation was on average 348 days (min 1 day; max 1,384 days; median 314 days) after the main investigation. For the purpose of this study, a total of $N = 568$ subjects underwent blood methylomic profiling (Data lock Apr. 2014). After pre-processing of methylomic data and genetic outliers exclusion, a total of $N = 533$ subjects were included in the methylomic profiling sample (Supplementary Table 1).

*Basel imaging sample.* This sample is part of the same ongoing, continuously recruiting imaging genetics study as the methylomic profiling sample. A total of $N = 753$ participants who were not part of the $N = 533$ methylomic profiling sample underwent imaging and EM assessment. A total of $N = 722$ subjects with complete imaging and EM assessment were included in the Basel imaging sample (Supplementary Table 1), among which $N = 623$ subjects underwent genotyping.

*Basel cognitive sample.* This sample is part of an ongoing, continuously recruiting genetics study in the city of Basel, Switzerland, independent from the methylomic profiling and Basel imaging samples. A total of $N = 1,622$ healthy young subjects underwent EM performance assessment and genotyping (mean age: 22.4; 66% female).

*Zurich sample.* This sample included a total of $N = 706$ healthy young subjects recruited in Zurich, who underwent EM assessment and genotyping (mean age: 21.8; 70% female).

All participants were free of any neurological or psychiatric illness, and did not take any medication at the time of the experiment (except hormonal contraceptives). The ethics committee of the Cantons of Zurich, Basel-Stadt and Basel-Landschaft approved the experiments. All participants received general information about the study and gave their written informed consent for participation.

*AgeCoDe sample.* This sample consisted of elderly participants of the German Study on Ageing, Cognition and Dementia in primary care patients (AgeCoDe). The AgeCoDe study is an ongoing primary care-based prospective longitudinal study on early detection of mild cognitive impairment and dementia established by the German Competence Network Dementia. The sampling frame and sample selection process of the AgeCoDe study have been described in detail previously[57] (see Supplementary Methods for complete description). Sufficient DNA-samples for genome-wide genotyping were available for 782 subjects. The complete description of EM phenotypes can be found in Supplementary Methods. The AgeCoDe-study was approved by the local ethic committees of all participating centres (Ethics Committee of the Medical Association Hamburg; Ethics Committee of the University of Bonn; Medical Ethics Committee II, University of Heidelberg at the University Medical Center of Mannheim; Ethics Committee at the Medical Center of the University of Leipzig; Ethics Committee of the Medical Faculty of the Heinrich-Heine-University Düsseldorf; Ethics Committee of the TUM School of Medicine, Munich). All participants received general information about the study and gave their written informed consent for participation.

*Munich sample.* The Munich sample consisted of patients with first episode and recurrent unipolar depression treated as in-patients at the Max Planck Institute of Psychiatry, Munich, and healthy control subjects ($N = 627$ with combined MRI and DNA availability; 423 patients, age 47.9 (s.d. 13.8) years; control subjects age 49.5 (s.d. 13.3) years), for the most part overlapping with imaging genetic and MDD association studies reported in collaboration with the ENIGMA consortium[58,59]. Other than in the flagship study[58], no bipolar patients were included for reasons of clinical homogeneity[59]. MDD diagnoses were based on clinical consensus in addition to M-CIDI or SCAN interviews, depending on the original study protocols. After pre-processing of methylomic data, and MRI-QC-based exclusions, combined data of $N = 596$ subjects was available for statistical analysis. Description of methylomic profiling and structural imaging of the Munich sample are provided in Supplementary Methods. All participants gave their written informed consent after receiving general information about the study. Study protocols and the transition of anonymous data into the biobank of the Max Planck Institute of Psychiatry were approved by the ethics committee of the Ludwig Maximilian University in Munich, Germany.

**Methylomic profiling.** Blood samples were collected from all the subjects using BD Vaccutainer Push Button blood collection set and 10.0 ml BD Vacutainer Plus plastic whole blood tube, BD Hemogard closure with spray-coated K2EDTA (Becton, Dickinson and Company, New Jersey, USA). DNA was isolated from the remaining fraction, upon plasma removal. The isolation was performed with QIAmp Blood Maxi Kit (Qiagen AG, Hilden, Germany), using the recommended spin protocol. Subject's DNA was extracted between midday and evening (mean time = 14:30, range 13:00–20.00). Microarray-based DNA methylomic profiling from whole-blood samples was performed at ServiceXS (ServiceXS B.V., Leiden, the Netherlands). In brief, the bisulfite conversion was performed with 500 ng genomic DNA input using the EZ DNA Methylation Gold Kit (Zymo Research, Irvine, CA, USA). A bisulfite conversion quality control on the samples was performed with DNA qPCR reaction and subsequent melting curve analysis[60]. The bisulfite-converted DNA was processed and hybridized to the HumanMethylation450 BeadChip (Illumina, Inc.), according to the manufacturer's

instructions. Methylation data were pre-processed using the R package RnBeads[61]. Beta values were calculated from SWAN normalized intensities[62]. Beta-values with detection $P$ value $\geq 0.05$ were considered as missing. Individual probes were excluded based on the following criteria: (1) non-CpG context probes, polymorphic probes, probes harbouring three or more SNPs in their 50mer extension (MAF $\geq 0.01$), and cross-hybridizing probes, based on the annotation provided with the RnBeads package[61], (2) cross-hybridizing probes and polymorphic CpGs sites referenced in refs 63, 64, (3) detected by iterative Greedycut algorithm, (4) missing rate $\geq 5\%$ in final samples. After quality control a total of 397,947 autosomal probes remained for analysis. Samples showing divergent genetic background from the majority of Caucasian samples were excluded; these genetic outliers were identified using Bayesian Clustering Algorithm[65] on genotypic projections onto the two first principal components inferred from reference Hapmap populations (CEU, JPT, CHB). Exclusion of samples yielded a total of 533 samples entering methylomic analyses.

To rule out systematic shift in DNA methylation values induced by SWAN normalization, we compared the correlation between summary statistics of CpG sites before and after normalization. We observed high correlation for both average ($r > 0.99$) and variance ($r > 0.95$) of DNA methylation values across samples. We also observed high average correlation between DNAm values before and after normalization per-CpG site (average $r = 0.87$), and per-sample (average $r = 0.89$ after mean-centring DNAm values per CpG).

DNA methylation profiles were obtained on average 1 year after imaging acquisition. We performed a sensitivity analysis examining the association between *ICA2* and cortical thickness after regressing out the difference ($\Delta$age) between age at blood sampling and age at MRI assessment from the methylomic pattern. The association remained significant ($P = 6.4 \times 10^{-5}$, $r = -0.18$ after adjustment for age) indicating that $\Delta$age did not affect the results of the study.

Primary phenotypes (age, cortical thickness, EM performance) were not confounded with methylomic processing covariates (plate, sentrix ID, position) (linear model minimum observed $P > 0.04$).

*ICA2* showed weak nominal association with Sentrix ID (Supplementary Data 6). After adjustment of *ICA2* for this technical covariate, the association with age, cortical thickness and EM performance remained highly significant (age: $P = 1.6 \times 10^{-11}$; age-adjusted thickness: $P = 1.3 \times 10^{-4}$, EM: $P = 0.0096$).

**Blood cell counting.** Haematological analysis, including blood cell counts, was performed at the collection time point with Sysmex pocH-100i Automated Hematology Analyzer (Sysmex Co, Kobe, Japan).

Lymphocytes, neutrophils and overall count of basophils, monocytes and eosinophils (mixture) were available for $N = 527$ participants from the methylomic profiling sample.

**Structural imaging.** Participants from the methylomic and Basel imaging samples underwent identical MRI assessment.

Measurements were performed on a Siemens Magnetom Verio 3T wholebody MR unit equipped with a 12-channel head coil. A high-resolution T1-weighted anatomical image was acquired using a magnetization prepared gradient echo sequence (MPRAGE) sequence with the following parameter: TE (echo time) = 3.37 ms, FOV (field of view) = 25.6 cm, acquisition matrix = $256 \times 256 \times 176$, voxel size = 1 mm $\times$ 1 mm $\times$ 1 mm. Using a midsaggital scout image, 176 contiguous axial slices were placed along the anterior $-$ posterior commissure (AC $-$ PC) plane covering the entire brain with a TR = 2,000 ms (flip angle = 8°).

From the initial $N = 533$ participants from the methylomic profiling sample and $N = 753$ from the Basel imaging sample, a total of 50 participants were excluded due to excessive movement or scanner noise by visual inspection of T1-weighted images, or technical reasons. This yielded a total of $N = 514$ from the methylomic profiling sample entering structural imaging analysis, and $N = 722$ subjects from the Basel imaging sample.

T1-weighted images were processed using the publicly available FreeSurfer software (v4.5) (refs 31–33). This processing includes motion correction, removal of nonbrain tissue, automated Talairach transformation, intensity correction, volumetric segmentation, and cortical surface reconstruction and parcellation. Specifically, the three-dimensional cortical surface was reconstructed to measure volume, surface area and thickness at each surface location or vertex. After the initial surface model was constructed, a refinement procedure was applied to obtain a triangulated representation of the grey/white (GM/WM) boundary. The GM/WM boundary was then deformed outwards to obtain an explicit representation of the pial surface. Thickness measurements were obtained by calculating the distance between the GM/WM boundary and pial surfaces at each vertex across the cortical mantle[31]. Global individual measures for thickness were computed by averaging cortical vertices measurements for both hemispheres. Individual measures were adjusted for sex, intra-cranial volume and MR-technical batches (software and gradient batches) using linear regression.

**ICA based identification of methylomic patterns.** After probes and samples quality control, missing Beta values were imputed using the R package impute. In order to adjust the methylation signals for technical confounders and preserve

effects of chronological age on methylation sites, we applied the iteratively re-weighted surrogate variable analysis algorithm implemented in the SVAR package[66], considering age at blood sampling as the outcome. Beta values were adjusted for sex and 40 inferred surrogate variables using linear regression. For each CpG, the residuals from this linear model were standardized across samples.

ICA decomposition of the standardized residuals was performed using the R package fastICA. The number of components to extract was estimated using the Random Matrix Theory algorithm[67] implemented in the R package isva[68]. Given the stochastic initialization of fastICA algorithm, we performed 30 repeats of the ICA components' estimation. All realizations of the mixing matrix ($A$) were clustered using hierarchical clustering, with complete linkage agglomeration, based on Pearson's correlation similarity. Final components were determined as the centrotypes of the inferred clusters.

When using such decomposition methods as ICA, multiple-correction depends on the number of identified components, which in not known a priori. In our case, the genome-wide methylomic data set was decomposed into 15 components that were amenable to downstream association testing. Hence, traits correlated with these 15 components were subjected to following α level adjustment: $P = 0.05/15 = 0.0033$. After having identified ICA2 as the only pattern associated with cortical thickness, we further investigated its relationship with eight regional thickness factor scores. The α level was thus adjusted for eight tests conducted ($P = 0.05/8 = 0.00625$).

**Association testing of methylomic patterns.** *Swiss sample.* The association between ICA patterns and imaging or behavioural phenotypes was assessed using Pearson's correlation, with two-sided association test. Given the delta between age at methylomic profiling and age at main investigation, chronological age adjustment was achieved by partialling out age effects: effect of age at methylomic profiling was regressed out from methylomic patterns and age at main investigation was regressed out from the phenotypic measure, using linear regression. Adjustment of methylomic patterns for blood cell counts was performed by regressing out effects of chronological age at blood sampling and effects of each of the three white-blood cell parameters using linear regression. The obtained residuals were subsequently tested for association with the relevant imaging or EM phenotypes.

Self-reported smoking frequency was measured on a 4-point Likert scale (0 = never, 1 = occasionally, 2 = 1–5 cigarettes per day, 3 = 6–20 cigarettes per day, 4 = 20 or more cigarettes per day). Self-reported alcohol consumption and cannabis use frequencies were measured on a 3-point Likert scale (0 = never, 1 = occasionally, 2 = daily). Association testing for each indicator was performed using linear regression.

*Munich sample.* ICA2 patterns were calculated separately for the whole Munich sample and a subsample of < 40-year-old subjects ($N = 163$). DNA methylation values were first adjusted for sex using linear regression. ICA2 patterns were then calculated as linear combination between the scaled residuals and the inverse ICA2 loadings inferred from the Swiss sample (Supplementary Data 8). Separate Pearson's correlation analyses were performed between ICA2-scores and biographical age, and ICA2-scores and cortical thickness. In addition, partial correlation analyses were performed between ICA2-scores and cortical thickness, correcting for age at MRI, difference between age at MRI and age at blood-drawing, sex, intracranial volume and MRI batch effects. All P values reported in the replication sample are one-sided.

**Exploratory factor analysis of cortical thickness measures.** Average regional cortical thickness in 68 areas (34 per hemisphere) were obtained from FreeSurfer automated parcellation method based on Desikan-Atlas[31–33]. Individual measures in each sample (methylomic profiling and Basel imaging samples) were adjusted for sex, intra-cranial volume, MR-technical batches and chronological age using linear regression. EFA was performed on regional cortical thickness measures from the methylomic profiling sample ($N = 514$). Factor extraction was based on principal axis factoring method. The number of factors to extract was determined using the parallel method implemented in R package psych. The factor analysis solution was rotated using the varimax method. A variable was considered to load on a factor if its absolute loading on the factor was 0.3 or greater. Based on the factor solution inferred from the methylomic profiling sample, we extracted factor scores predictions for both the methylomic profiling sample and the independent Basel imaging sample, using regression method. For each sample, factor scores were tested for association with EM performance using Pearson's correlation, with a two-sided association test. The same association analysis was conducted combining factor scores and EM performance of the two samples (combined sample, $N = 1,234$).

**Mediation analysis.** Chronological age at main investigation, ICA2 methylomic pattern and global cortical thickness were entered in a mediation analysis[28] using the R package MBESS. To represent the strength of the mediation we computed the indirect effect ($a$ multiplied by $b$, see Fig. 1) and the $k^2$ square value which is interpreted as the proportion of the maximum possible indirect effect that could have occurred[69]. The 99.9% confidence intervals for these parameters were obtained on bias-corrected and accelerated bootstrapping procedure with 10,000 resamplings. Significance of the indirect effect was assessed by testing whether the

confidence interval of the indirect effect excludes 0, considering interval limits from 90 to 99.9%.

**Gene-set enrichment analysis of methylomic component.** CpGs were mapped to transcripts based on Illumina's annotation; EntrezID gene identifiers were downloaded from the UCSC genome database. Enrichment testing was performed using the GOseq package[40] which applies stringent correction towards genes mapped by multiple CpGs across the array. Enrichment statistics were obtained using Wallenius approximation. A total of 19,518 genes mapped by the 397,947 CpGs entering the analysis were used as background. Gene sets were downloaded from the MSig DB (www.broadinstitute.org/gsea/msigdb/, curated gene lists C2 and C5).

**Genetic association analyses.** *Genotyping.* DNA was extracted from saliva or blood using standard protocols. All subjects were individually genotyped using the Affymetrix Human SNP Assay 6.0 according to the manufacturer's recommendation. In the methylomic profiling sample, subjects with unusual ancestry according to the majority of the sample were excluded using Bayesian clustering algorithm and Hapmap reference populations (see Methylomic profiling). Subjects were also checked for inconsistency between reported and genetically inferred sex. Individual call rate averaged to 98.3%. For the purpose of scoring analyses, subjects from the Basel imaging, Basel cognitive, Zurich and AgeCode samples were additionally excluded based on the following criteria: genome-wide call rate < 95%; IBD sharing defined by PI_HAT > 0.2 (one-sample of each detected pair was excluded); Bayesian Clustering[65] outlier detection on genome-wide call rate and heterozygosity rate. This yielded a total of $N = 1,445$ individuals entering the genetic scoring analysis for Basel cognitive sample, $N = 534$ for Basel imaging sample, $N = 624$ for Zurich sample and $N = 743$ for AgeCode sample.

*Validation of the link between methylation and genotype data.* A per-subject crosscheck between phenotypic data, methylation data and genetic data was performed using the reported sex and sex-predictions based on the array data, as well as matching of all SNPs represented on the Illumina 450 K array to the corresponding Affymetrix SNP 6.0 genotype calls. This crosscheck allowed an unambiguous assignment of each methylation data set to the corresponding genetic and phenotypic data set.

*Gene set enrichment analysis of ICA2 pattern.* GSEA was performed using the MAGENTA[44] software which derives gene-centric association statistics from single-SNP association P values, while controlling for potential confounders (gene size, number of SNPs, number of independent SNPs, number of recombination hotspots, linkage disequilibrium and genetic distance). Genome-wide single-SNP association analysis was conducted on ICA2 pattern adjusted for chronological age (at blood sampling), using an additive model. A total of 773,330 autosomal SNPs that passed individual SNP quality control in the methylomic profiling sample (exclusion criteria MAF < 0.01; HWE P value ≤ 0.0001; call-rate < 0.90) entered the analysis.

In order to capture signals from potentially regulatory variants, MAGENTA-derived gene scores were based on SNPs lying within 20 kb upstream and downstream of the extreme transcript boundaries. The GSEA algorithm includes a built-in procedure controlling for physical proximity of SNPs within a given gene set (automatic exclusion of the gene exhibiting a lower association signal in case of one SNP mapped to multiple genes within a gene set). Gene set enrichment statistic was based on the 75th percentile cut-off of the observed genome-wide gene-score distribution, which has been proposed to show optimal power for weak genetic effects as expected for complex polygenic traits. Empirical P values were adjusted for multiple testing using FDR. Gene sets were extracted from the MSigDB v3.1 database (http://www.broadinstitute.org/gsea/msigdb), including gene sets from different online databases (KEGG, Gene Ontology GO, BioCarta and Reactome). We used a gene set size ranging between 20 and 200 genes to avoid both overly narrow and broad gene set categories, resulting in 1,411 gene sets to be analysed. Genes from the extended major histocompatibility complex region were excluded from the analysis.

*Genetic scoring association analyses.* The scores comprised SNPs associated with ICA2 pattern ($P < 0.05$) mapping to an equal number of genes (that is, one most significant SNP per gene). Genetic scores were computed using the PLINK[70] score profile procedure. Scores were weighted by the direction of effect of association ( + 1 or − 1) of each minor allele with ICA2 pattern inferred from the methylomic profiling sample. Genetic score calculations were restricted to SNPs meeting the inclusion criteria: MAF ≥ 0.01; HWE P > 0.0001; call-rate ≥ 90% within a sample. Resulting genetic profiles were adjusted by regressing out the effect of the number of missing SNPs per-subject included in the scoring procedure. Associations between the inferred scores and behavioural or imaging phenotypes were assessed using Pearson's correlation. Given the negative correlation observed between ICA2 methylomic pattern and EM performance in the methylomic sample, genetic score correlation tests with EM performance were one-sided (lower tail).

Description of additional analyses of ICA methylomic patterns can be found in Supplementary Methods.

**Data availability.** The data that support the findings of this study are available from the corresponding authors on request.

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

## Acknowledgements

This work was funded by the Swiss National Science Foundation (Sinergia grant CRSI33_130080 to D.J.F.d.Q. and A.P., individual grant 320030_159740 to D.J.F.d.Q.), by the European Community's Seventh Framework Programme (FP7/2007–2013) under grant agreement #602450 (IMAGEMEND), by the German Research Network on Dementia (KND) and the German Research Network on Degenerative Dementia (KNDD), German Federal Ministry of Education and Research grants 01GI0420 and 01GI0711, and by the University of Basel. We thank Elmar Merkle, Christoph Stippich and Oliver Bieri for granting access to the MRI facilities of the University Hospital Basel. Preprocessing of the brain imaging data was performed at sciCORE (http://scicore.unibas.ch/), the centre for scientific computing at the University of Basel.

The Munich sample comprised images acquired in subsamples of the Munich Antidepressant Response Signature Study and the Recurrent Unipolar Depression Case-Control study, both performed at the MPIP. We thank radiographers Rosa Schirmer, Elke Schreiter and Ines Eidner for image acquisition, data organization and preparation. The studies have been supported by a grant of the Exzellenz-Stiftung of the Max Planck Society and also been funded by the Federal Ministry of Education and Research (BMBF) in the framework of the National Genome Research Network (NGFN), FKZ 01GS0481.

## Author contributions

V.F., A.P. and D.J.F.d.Q. conceived and designed the study. V.F., T.C.-R., A.M., P.G.S., V.V., D.C., P.D., T.E., L.G., E.L., C.V., D.J.F.d.Q. and A.P. analysed the data. F.J., W.M., S.G.R.-H., M.S., M.W. and E.B.B. provided samples and analysed the data. A.P., V.F. and D.J.F.d.Q. wrote the manuscript. All collaborators reviewed and approved the final manuscript.

## Additional information

**Competing interests:** The authors declare no competing financial interests.

