## [Peer Review File · Nature Communications]

Reviewers' comments:

Reviewer #1 (Remarks to the Author):

In this manuscript, the authors describe a peripheral blood DNA methylation signature that correlates with cortical thickness, especially of temporal regions as well as memory performance. The authors next identified genetic variations that associate with the epigenetic signature and from this identified multilocus SNP sets that predicted memory performance in several samples. Both the DNA methylation and SNP sets pointed to an overrepresentation of genes within immune pathways.

The approach is original, as it is the first to identify peripheral blood epigenetic signatures to correlate both with structural brain imaging data as well as neurocognitive performance. The identification and replication in multiple large samples is another positive aspect.

The authors use ICA to identify patterns of DNA methylation. These are then correlated first with whole brain cortical volume. Here it is not clear how robustly the ICA would pick the same patterns in a second sample. This is maybe important, since the authors functionally annotated the features in the ICA.

Furthermore, the authors correct for cell count in the first sample using actual blood cell counts. While this is laudable, important distinctions among subclasses, such as T-cells cannot be made. Bioinformatic annotation of blood cell types could help to identify whether the signature may in fact be derived more from one or the other blood cell type that may not have been measured.

The authors identify ICA2 which is associated with cortical thickness even after correcting for age, while still remaining correlated with age. Did the authors test whether these CpGs contain some of the known epigenetic clock CpGs (Horvath, Hanon?).

The authors then test whether the ICA associates with genetic variation using GSEA. Here two GO terms are enriched. The weighted scores from the genetic associations associate with memory performance in 5 samples. It is not clear whether this is not also a circular approach, as a number of papers describe that variance in DNA methylation is often carried by underlying genetic variation. Would a combined score from genetic and epigenome analysis perform even better? Are the SNPs enriched in the GSEA within mQTLs for the CpGs in the ICA2?

Overall this is an interesting paper with some robust and replicated findings. Some questions regarding the robustness of the ICA remain open and the relevance of the genetic analysis. This reviewer is not a statistics expert and would defer comments regarding ICA to reviewers more expert in the field.

Reviewer #2 (Remarks to the Author):

This study tests to what degree methylomic profiles can account for age-related cortical thickness decrease. They found a significant contribution from epigenetic marks, which further point to genes involved in immune system regulation. There were also tendencies that the cortical region where thickness was related to this marker correlated with episodic memory performance. I think this is a well-conducted and interesting study, which adds to our understanding of the very pervasive phenomenon of cortical thinning with age. The study is original and will be of great interest to the research community.

I have a few questions and suggestions, especially regarding some of the analyses presented and how the sample(s) are described.

ICV and thickness are usually not highly correlated. To remove global correlations prior to EFA, maybe adjustment for mean thickness would be more appropriate?

The best strategy to handle the common influence of age, while not throwing the baby out with the water, is challenging. I think the mediation analysis used by the authors is a way of showing the mutual relationship between the variables in a clear way. In the EFA analyses, however, age was regressed out prior to the factor extraction. That may be ok, but it was not entirely clear to me how age was handled in the following analyses. I understand that ICA2 patterns were used, but were the age-adjusted thickness estimates also used, or rather the raw thickness estimates based on the age-adjusted factors? If the latter is the case, maybe this could be followed up by a mediation analysis where age is included, since it is a major source of variation that is now hidden. As I read the rationale for the study, understanding age-related changes in cortical thickness is a major objective, and it would thus be informative if age was not just taken out of the equation. Similar concerns regard the memory analyses.

It is mentioned that synaptic pruning during development may account for cortical thinning. It could also be mentioned that myelination of lower cortical layers could cause the cortical mantle to appear thinner on the MR scans. This likely accounts for a substantial part of the observed cortical thinning in development.

Could the presentation of which samples that were used for the different analyses be more clearly presented, e.g. in a table, figure or flow chart? If I understand correctly, the age-thickness-methylome associations were found in a sample of your participants only? This should be more clearly stated in the ms, with age-ranges also. One easily gets the impression that this is an older-age study also, and I think this needs to be much more directly addressed, and also touched upon in the discussion, because it will have substantial impact on how the results are conceived.

Minor point:

Is it possible to include some quantitative information in the abstract also, such as sample size and effect sizes?

The Fischl et al paper on whole-brain segmentation would not be the most appropriate as the first FreeSurfer-reference here, since the present paper is focused on cortical thickness and the Fischl paper on volume. The 1999 and 2000-papers from Dale and Fischl should rather be referenced first instead.

Reviewer #3 (Remarks to the Author):

Freytag and colleagues analysed whole blood DNA methylation profiles generated using the 450k array in ~500 individuals and correlated these (primarily) to cortical thickness. The authors employed a wide collection of statistical techniques including state-of-the-art approaches. However, the analysis largely skips accepted straight-forward approaches for the analysis of DNA methylation data in population studies and instead heavily relies on 'black-box' types of statistical analysis, more suitable for second order analyses. It therefore is difficult to assess what data go into the analysis and even more so to interpret the outcome. It therefore remains questionable whether this study, as the authors claim, shows that the decomposition of blood methylomes bears considerable potential for the study of brain-related traits.

- The straight-forward analysis that should be presented is an epigenome-wide association analysis (EWAS) testing for association between the methylation of individual CpGs and cortical thickness. This approach allows the reader to gauge the relationship between the 'raw' data and the outcomes.
- The authors instead use independent component analysis (ICA) and test the resulting components, without explaining why they jumped to such a global analysis and why specifically ICA. It seems ICA is an alternative to the more common principle component analysis (PCA) when the requirements for PCA are not met: the data are non-Gaussian and (very) noisy. The authors do not state why they used ICA but show data that indicate that the data are indeed quite noisy which raises doubts about the validity of the data sets (an may preclude an EWAS-type analysis): (1) ICA decomposition resulted in 126 independent component, 111 (88%) of which were driven by a single individual. This suggests that 111 of the 533 individuals actually represent outliers perhaps due to technical issues and probably should have been omitted from the analysis. (2) SVA identified 40 surrogate variables which is very large considering the sample size and indicates significant technical variability.
- Since the primary research question relates to the association of whole blood DNA methylation profiles with cortical thickness, SVA should be performed using cortical thickness as outcome, not age. Now, the interpretation of associations of ICA components remains unclear: is it driven by age, an age-effect on cell composition etc?
- If the primary research question was to gain insight in the role of age in the decline of cortical thickness (as perhaps suggested in the introduction), a specific analysis of known age-associated CpGs (or scores based on multiple CpGs) in blood would have been in order (as mentioned in introduction).
- The 15 ICAs not driven by a single individual are not described. How much of the variability do they explain? What is their nature: with which (technical, cell count variable etc) variables do they correlate? It is elemental to know which CpGs contributed to the ICAs to contribute to our biological understanding. Are they known to be associated with age or differentially methylated between fine grained cell counts?
- The main PCAs in whole blood 450k data usually capture cell counts (beyond the larger classes like the 3 measured here). It is not unlikely that the ICAs also represent cell counts. Hence, if true, the data may suggest an association between the immune system (measured through DNA methylation instead of direct cell counts) and cortical thickness. If this association survives robust control for confounding (i.e. including parameters known to be associated with cortical thickness (and immune system traits) (social economic position, medication use, etc.)). This in fact may be a very interesting observation but requires much more specific insight in the biology of the ICAs (if any).
- Did the authors validate the link between methylation and genotype data (e.g. were they from the same person)?
- Using genetic variation as causal anchor is a strong approach. However, identifying robust genetic variants requires a GWAS and hence many thousands if not 10 thousands of samples (instead of 500). Were there GWASs performed for cortical thickness the authors can rely on? Gene-set enrichment approaches as applied here reduce the enthusiasm for a genetic approach and limits its interpretation.
- SWAN sometimes induces strange DNA methylation values. It will be useful to exclude such artefacts by comparing the raw DNA methylation values to those post-SWAN.
- How many methylation values were missing and imputed? What were setting of the impute package?
- DNA methylation data seems to be generated on a sample taken ~1 years after the MRI. It should be discussed how this affected the study.
- Why were the measured cell counts not included in SVA?

Reviewers' comments:

Reviewer #1 (Remarks to the Author):

The authors have added additional analyses to address all remaining concerns.

Reviewer #2 (Remarks to the Author):

The authors have address all my initial concerns.

Reviewer #3 (Remarks to the Author):

I genuinely appreciate the considerable additional work the authors put into the manuscript. I believe this has strengthened the manuscript and improved the clarity of their reasoning. I feel, however, that when adopting non-standard analysis approaches (ICA instead of EWAS) particularly in a novel field like epigenetic epidemiology, the burden of proof that the outcomes are robust is on the authors. I have three major concerns, of which certainly the latter two can be addressed easily.

1. The finding that ICA2 is associated with cortical thickness is based on a single study of 553 individuals. Replication is a key aspect of any genomic association study.
2. The authors claim that the association of ICA2 with cortical thickness and other phenotypes is not driven by blood cell counts. In general, cell counts are well-known to be main drivers of major components explaining methylomic variance. Accounting for main cell types is not sufficient to exclude this possibility. The fact that ICA2-CpGs are near genes that are primarily involved in processes related to inflammation and leukocyte differentiation is compatible with a cell count effect. As mentioned in my first review, an immune component related to cortical thickness is of interest but we should know its nature. The authors may use the approach adopted in a recent paper on age-related changes in DNA methylation (Slieker et al. *Genome Biol* 2016; 17:191), where the authors took public 450k data from many blood cell subtypes and then assess whether identified CpGs are differentially methylated between cell subtypes. The authors can do this for their 970 ICA2 CpGs.
3. The authors do use a public data set (Hannum et al) to confirm the presence of ICA1 and 2 and their correlation with age. This data-set was, however, on whole blood. They should extend this analysis to public data on purified blood cells like the one published in *Nat Commun* covering monocytes and T cells (Reynolds et al. *Nat Commun*. 2014; 5:1–8). This would further substantiate the independence of blood cell counts.

Reviewers' comments:

Reviewer #3 (Remarks to the Author):

I wish to laud the authors for their additional work. Seeking and finding replication (despite the risk of falsifying their original finding) and a considerable effort to gain more insight in the impact of blood cell composition very much strengthens the manuscript. The results are intriguing and accordingly the authors present a cautious discussion on the explanation for and the implications of their findings. So, I am happy with the revisions.

Reviewer #4 (Remarks to the Author):

Freytag et al. identified an epigenetic signature that was associated with cortical thickness and memory performance in 533 healthy young adults. The epigenetic signature was replicated in 596 participants with major depressive disorder and healthy controls. Further, the authors showed that this signature mediated the effect of age on cortical thickness. I consider the findings novel and the application of ICA as analytical strategy of interest to the community.

As advised by the Editor, I have specifically focused this review on the aspects of the study that related to the DNA methylation analysis. I fully share the initial concerns raised by Reviewer 3 that: (a) replication of the reported associations in an independent sample cohort is critical; and (b) the reported associations need to be independent of differential blood cell counts.

Re (a): I acknowledge the effort that was necessary to add these data and analyses to the revised manuscript. The authors replicated their primary finding, and therefore have addressed my concerns in this regard.

Re (b): The authors performed the additional analyses requested by the reviewer. The results showed that after adjustment for differential cell counts, the associations of ICA2 with both chronological age and cortical thickness remained significant. Further, the authors showed that the ICA-age correlations identified in whole-blood were also detectable in individual cell types. I note that the authors have added a paragraph to the Discussion section (p. 16), indicating the caveats of DNA methylation profiling in a heterogeneous tissue such as peripheral whole blood. Together, I think the authors have done an adequate job in replicating their findings and assessing confounding due to cellular composition.

Nonetheless, it would be useful if the authors considered the below concerns, suggestions or additions in the revised manuscript:

(1) A justification as to why ICA was applied in favour of other more established analytical approaches could be added. In addition, the authors could elaborate on the limitations of ICA in the Discussion section.

(2) It would be important to add a clarification on how significance thresholds were defined, particularly for the association testing between the ICA components and traits of interest. Is there an appropriate equivalent of 'genome-wide significance'?

(3) Related to (2): Some reported correlations have highly significant p-values ($p < 2.2e-16$), yet the corresponding correlation coefficients seem low ($r = 0.32$). Some associations are dismissed as speculation, despite reporting $p < 1e-60$ as for the case of whole blood vs. monocyte ICA2 DNAm

values. I am not challenging the statistics, but the authors could make a better effort in reporting the statistics more consistently and with more caution throughout the manuscript. After all, insight into molecular mechanism is not provided.

(4) Supplementary Figure 1: I assume the y-axis indicates proportion and not percentage?

(5) Introduction: "[...] a single nucleotide polymorphism (SNP)-based, genome-wide study of methylome patterns' genetic variability". The sentence structure could be improved.

Reviewer #5 (Remarks to the Author):

Increasing age is tightly linked to decreased thickness of the human neocortex. The authors identified an epigenetic signature that was associated with cortical thickness in 533 healthy young adults and replicated this finding in 596 participants with major depressive disorder and healthy controls. The authors reported that the epigenetic signature mediated the effect of age on cortical thickness and pointed to the involvement of immune system genes.

I believe the findings are robust but have several concerns about the interpretation of the results.

1. Two out of 15 ICA methylomic patterns (termed ICA1 and ICA2) were significantly correlated with age but only ICA2 predicted cortical thickness after controlling for the linear effects of age.

Supplemental table 2 suggests that ICA1 captures the linear affects of age. My question is whether ICA2 captures non-linear components of the effects of age. For this purpose, I propose an analyses where cortical thickness is first predicted from a model containing a 5th degree polynomial of age (i.e., $\text{age} + \text{age}^2 + \dots + \text{age}^5$). If ICA2 is added to this model, does it significantly increase the overall R^2 ?

I believe, this answer to this question is important because if this test is not significant, the relation between methylation and cortical thickness could be spurious and caused by age affected both cortical thickness and methylation profiles in blood.

2. The authors claim that the epigenetic signature of age on cortical thickness. ($p < 0.001$). This seems too strong given the non-experimental nature of the data and possibility of alternative explanations. A mediator model predicts that the age-cortical thickness correlation equals the product of the age-epigenetic signature correlation times the epigenetic signature-cortical thickness correlation. This model does not seem to hold making the authors to conclude that the epigenetic signature only partly mediated the affects of age. However, I am unsure about this statement, as this is just a decomposition of a correlation and alternative explanations for such data that that do not assume mediation. (e.g., there could be 3rd variable affecting both cortical thickness and methylation).

3. I wonder if other phenotypic information is available to further study ICA2. Examples include information on physical exercise, smoking, diet, health indicators etc. Currently, the interpretation of ICA2 relies heavily on bioinformatics and blood methylation studies of implicate immune system genes, making this result a bit generic.

4. The authors state that DNA methylation age values were significantly correlated with actual participants' age. As DNA methylation age is essentially the deviation from methylation predicted age and chronological age, this should not be the case. It makes me wonder whether this is caused by

possible non-linear effects of age remaining in the DNA methylation age values .

5. A multilocus genetic score reflecting genetic variability of this signature was associated with memory performance ($p=0.0003$) in 3346 young and elderly healthy adults. Was this multilocus score also correlated with cortical thickness?

6. Previous studies have linked immune system to cortical thickness. I wonder if the authors could speculate a little about how this methylation profile might be useful (if not simply a non-linear effect of age) could further advance the study of (age-related) cognitive decline.

REVIEWERS' COMMENTS:

Reviewer #4 (Remarks to the Author):

The authors have satisfactorily addressed my concerns, and have provided additional clarification where necessary.

Reviewer #5 (Remarks to the Author):

My main concern was that the observed relation between cortical thickness merely represented non-linear effects of age. The additional analyses showed this was not the case.

I now believe that the claim from the authors that they found a methylation signature that is linked to the immune system and predicts cortical thickness is reasonable.

There are multiple other reports linking the immune system to cortical thickness so the novelty of the paper is that rather than traditional immune system markers it suggest the possibility of using a methylation signature.

I still think the authors could have provided a more extensive discussion of the possible advantages of the methylation signature over other immune system markers. For example, traditional immune system markers may reflect the present state, it is very well possible that the methylation preserved a record of past immune response. Further, it may provide a more powerful marker than SNPs due to higher correlations with cortical thickness. I guess such a discussion would serve to point out the unique value of their findings and contribution to the existing literature.

Reply to Reviewer #1

We are grateful to this reviewer for the constructive and helpful remarks. In order to address these, we studied additional samples and performed new analyses.

Reviewer #1, comment 1:

The authors use ICA to identify pattern of DNA methylation. These are then correlated first with whole brain cortical volume. Here it is not clear how robustly the ICA would pick the same patterns in a second sample. This is maybe important, since the authors functionally annotated the features in the ICA.

Authors' response:

This comment motivated us to include an independent sample and test the between-sample overlap of ICA methylation patterns. To this end, we performed ICA analysis (for detailed description, see Methods below attached to this response) of the study population reported in Hannum et al., 2013 (Hannum G. et. al. Genome-wide methylation profiles reveal quantitative views of human aging rate. *Mol Cell*. **49**, 359-67. (2013)), which consists of n=656 blood DNAm profiles of participants spanning a wide age-range (19-101, mean age: 64 years).

In the first step of the analysis we identified 5 ICA patterns that were significantly associated with age (designated HICa-e, see Table 1 in this response). In the second step we examined the overlap between *ICA1* and *ICA2* CpGs identified in our sample of healthy young adults and CpGs contributing to each of the Hannum age-associated IC pattern. A significant overlap with *ICA1* was observed for one pattern (OR=90.9, $p < 1e-60$, see Table 1 in this response). For *ICA2* we observed a significant overlap with three age-associated Hannum patterns (see Table 1 in this response), with said overlap being particularly strong for Hannums' HICd pattern (OR=48.7, $p < 1e-60$). Importantly, the correlation of loadings between CpGs contributing to *ICA2* and CpGs contributing to Hannums' HICd pattern was positive and substantial ($r=0.87$, $p < 1e-60$).

Thus, we observed highly significant between-sample overlap of ICA patterns despite the differences in age structure of these samples, supporting the robustness of the methods presented herein. We added this data on pages 7 and 22 of the revised manuscript.

Table 1: Age-associated ICA DNAm patterns in the Hannum et al study.

Hannum's IC	Association between age and IC pattern		# CpGs (a)	CpGs Loadings overlap for ICA1		CpGs Loadings overlap for ICA2	
	r	p		OR (b)	p (c)	OR (b)	p (c)
HICa	0.68	<1e-60	2493	90.9	<1e-60	3.5	1.9e-6
HICb	0.35	<1e-60	3236	0	1	0.12	0.99
HICc	0.26	1.7e-11	2142	0.25	0.98	9.4	2.4-e29
HICd	0.17	1.3e-05	1766	0.60	0.83	48.7	<1e-60
HICe	-0.16	4.3e-5	3210	0.17	0.99	0	1

(a): Number of contributing CpGs for Hannum's age associated IC.

(b): Odds-ratio for a fisher's exact test testing enrichment of ICA1/ICA2 CpGs in Hannum's signature; (c): p-value for fisher's exact test.

Methods related to this response:

We analyzed whole-blood methylomic profiles from n=656 samples published in Hannum et al., 2013. In analogy to our methylomic dataset, multi-mapping or polymorphic probes were excluded from analysis. Raw intensities (methylated and unmethylated signals) were normalized using the lumi package (color-bias adjustment and quantile normalisation). The BMIQ algorithm was finally applied to adjust for the difference between Type I and Type II probes used in the 450K array. Given substantial non-randomness of between-plate distribution of chronological age in this sample, we performed CoMbat adjustment for plate effect. DNA methylation values were subsequently adjusted for sex and 98 surrogate variables inferred from surrogate variable analysis (SVA).

ICA decomposition on the adjusted signals yielded a total of 175 components, among which 19 were retained based on the per-subject 10% variance criterion used in our methylomic dataset. The retained ICA patterns were tested for association with age, after adjustment for estimated cell counts (CD4T, CD8T, NK, Gran, Mono, Bcell). Five patterns were significantly associated with age. In analogy to our study, CpGs contributing to these patterns were chosen so as to exhibit an absolute loading > |4| on the respective pattern.

Reviewer #1, comment 2:

Furthermore, the authors correct for cell count in the first sample using actual blood cell counts. While this is laudable, important distinctions among subclasses, such as T-cells cannot be made. Bioinformatic annotation of blood cell types could help to identify whether the signature may in fact be derived more from one or the other blood cell type that may not have been measured.

Authors' response:

We followed the reviewer's suggestion and used bioinformatic annotation of blood cell types as described by Jaffe & Irizarry (Jaffe and Irizarry, 2014: Accounting for cellular heterogeneity is critical in epigenome-wide association studies. *Genome Biol.* 15:R31) After this adjustment, *ICA2* associations with both chronological age and cortical thickness remained highly significant (age: $r=0.29$, $p=2e-11$; cortical thickness: $r=-0.22$, $p=8.3e-7$). We added this data on page 7 of the revised manuscript.

Methods related to this response:

Blood sub-cell types proportion (CD4T, CD8T, NK, Gran, Mono, B cell) were derived from DNAm signal using Jaffe's and Irizarry's method. Adjustment for association testing was performed by regressing out effects of the six estimated proportions from the *ICA2* pattern using linear regression.

Reviewer #1, comment 3:

The authors identify ICA2 which is associated with cortical thickness even after correcting for age, while still remaining correlated with age. Did the authors test whether these CpGs contain some of the known epigenetic clock CpGs (Horvath, Hannum?).

Authors' response:

To address the reviewer's comment, we first applied Horvath's cross-tissue- and Hannum's whole-blood-based predictor in our sample. Both predictors yielded DNA methylation age values that significantly correlated with actual participants' age (Horvath's predictor: $r=0.70$, $p < 1e-60$; Hannum's predictor: $r=0.71$, $p < 1e-60$). Among the 970 CpGs constituting *ICA2*, one marker (cg18055007) was part of the 353 Horvath

predictor markers, and four (cg20822990, cg16054275, cg16867657, cg21139312) were part of the 71 CpGs included in Hannum's DNAm age model. Neither predictor was associated with cortical thickness after adjustment for chronological age (Horvath's: $r=0.04$, $p=0.32$; Hannum's: $r=0.01$, $p=0.77$), suggesting that these predictors (like *ICA1* but, importantly, unlike *ICA2*) do not mediate the effect of age on cortical thickness. We added this information on page 7 of the revised manuscript.

References related to this response:

Hannum G et al. Genome-wide methylation profiles reveal quantitative views of human aging rates. *Mol Cell*. 2013 Jan 24;49(2):359-67.

Horvath S. DNA methylation age of human tissues and cell types. *Genome Biol*. 2013;14(10):R115.

Reviewer #1, comment 4:

The authors then test whether the ICA associates with genetic variation using GSEA. Here two GO terms are enriched. The weighted scores from the genetic associations associates with memory performance in 5 samples. It is not clear whether this is not also a circular approach, as a number of papers describe that variance in DNA methylation is often carried by underlying genetic variation. Would a combined score from genetic and epigenome analysis perform even better?

Authors' response:

We thank the reviewer for the opportunity to improve clarity regarding this point in the paper:

The reasons to perform a genetic association study in different samples were two-fold: 1) Given the fact that genetic variation is indeed related to DNA methylation (as correctly pointed out by the reviewer), we tested whether *ICA2*-related genetic variation can be used as a proxy for DNA methylation in larger samples, where such epigenetic measures were unavailable. This was indeed the case. 2) We hypothesized that the biological processes revealed through gene-set enrichment would be similar regardless of the nature of the data input (i.e. genetic vs. epigenetic variation). This was also the case and increased the confidence in non-randomness of the results presented herein.

We added this information on pages 15 and 16 of the revised manuscript.

With regard to a combined genetic-epigenetic analysis, it would be only possible in our discovery sample, however, such an analysis would lead to massive inflation of type I error due to model overfitting. As pointed out above, such an analysis was not possible in the additional samples, because for these only genetic data was available.

Reviewer #1, comment 5:

Are the SNPs enriched in the GSEA within mQTLs for the CpGs in the ICA2?

Authors' response:

Addressing this point required multiple analytical steps:

First, we examined the location of the 71 GSEA-selected genetic score SNPs (39 SNPs for *Hemopoietic or lymphoid organ development* and 33 for *Pyk2/Keratinocyte pathway*, one SNP common to both gene sets) with regard to the 970 CpGs constituting *ICA2*. We observed a significant over-representation (53%, $p < 2e-4$) of gene score SNPs *in cis* (i.e., ± 1 Mbp) to *ICA2* CpGs as compared to a genome-wide random distribution) (see Methods attached to this response).

Next, we performed mQTL analysis for each of the gene score SNP - *ICA2* CpG pairs. Examination of the QQ-plot of association statistics demonstrated significant deviation from the null uniform distribution with particular over-representation of multiple genetic associations with effect sizes ranging from small to moderate (see Figure 1 attached to this response). Thus, GSEA-derived SNPs collectively exert multiple genetic effects of small to moderate magnitude on the CpGs contributing to *ICA2*. We added this data on pages 13 and 27 of the revised manuscript.

Methods related to this response:

Testing over-representation of genetic score SNPs in -cis to ICA2 CpGs: We randomly selected an equal number of SNPs from genome-wide genotyped SNPs and assessed the occurrence of SNPs found in -cis (± 1 Mbp) to *ICA2* CpGs. This sampling procedure was repeated 5000 times to establish the null distribution and calculate the corresponding p -value.

Null distribution of cis-mQTL association statistics: first we determined all SNPs located within ± 1 Mbp of any of the *ICA2* CpGs ('cis-SNP pool'). Association statistics were computed between *ICA2* CpGs and n SNPs randomly selected from the cis-SNP pool, with n equal to the number of GSEA genetic score SNPs (i.e. 71 SNPs), thus providing one realization of the baseline quantile distribution. This sampling procedure was repeated 1000 times.

Figure 1: Q-Q plot of mQTL analysis between 71 GSEA genetic score SNPs and *ICA2* CpGs.

Red line shows expected uniform distribution.

Blue dashed line indicates the 95 % quantiles obtained from 1000 repeats of association testing between *ICA2* CpGs and randomly selected cis-SNPs.

Reply to Reviewer #2

We highly appreciate the constructive remarks made by this reviewer.

Reviewer #2, comment 1:

ICV and thickness are usually not highly correlated. To remove global correlations prior to EFA, maybe adjustment for mean thickness would be more appropriate?

Authors' response:

Adjustment for covariates prior to exploratory factor analysis (EFA) aimed at removing global correlations driven by factors not of interest in the present analysis (e.g. age, ICV, sex, technical batches). It is true that adjustment for ICV did not impact on the EFA solution (correlation between factors' loadings with/without adjustment for ICV: > 0.99). Yet, we believe that adjustment for mean thickness would not be appropriate for the aim of the present analysis: given the strong association between *ICA2* and global thickness, such adjustment would exclude an important fraction of variance possibly related to *ICA2*, thus reducing the power of detecting an association between the methylomic pattern and the EFA extracted patterns.

Nevertheless, in order to address this point, we performed EFA under adjustment of the ROIs for mean thickness (and not for ICV). We observed a high mean correlation between factor loadings across the two EFA solutions ($r = 0.78$); importantly, *F6* remained stable across the two solutions with an $r = 0.89$ ($p = 6.9e-24$) between loadings before/after adjustment for mean thickness)(see Table 1 in this response), suggesting that the results presented herein were not driven by global mean thickness.

In addition, *F6* scores obtained from the mean thickness-adjusted EFA solution were still significantly associated with EM performance ($p = 0.008$) and *ICA2* ($p = 0.042$) (see Table 2 in this response).

We added this information on page 9 of the revised manuscript.

Table 1: Factor Loadings correlations before/after adjustment for mean thickness.

Thickness adjusted EFA (a)	ICV adjusted EFA (b)	Loadings correlation	
		r	p
F1	F1	-0.70	4.6e-11
F2	F2	0.84	2.3e-19
F3	F3	0.81	7.8e-17
F4	F7	0.71	1.2e-11
F5	F4	0.94	1.0e-31
F6	F6	0.89	6.9e-24
F7	F5	0.94	1.2e-33
F8	F8	-0.45	1.e-04

For each factor from the mean thickness-adjusted EFA (a), the factor from the ICV adjusted EFA (b) exhibiting highest loadings correlation is shown.

Table 2: Association between factor scores from mean thickness adjusted EFA and ICA2/EM performance.

Factor	ICA2 (N=514)		EM (N=1234)	
	r	p	r	p
F1 (F1)	0.019	0.67	0.047	0.098
F2 (F2)	-0.001	0.99	-0.002	0.95
F3 (F3)	-0.002	0.96	0	0.99
F4 (F7)	0.022	0.62	-0.021	0.46
F5 (F4)	-0.018	0.69	-0.014	0.62
F6 (F6)	-0.09	0.042*	0.075	0.008*
F7 (F5)	0.041	0.35	-0.035	0.22
F8 (F8)	-0.019	0.67	0.058	0.043

In brackets: factor from the unadjusted mean thickness EFA showing the highest loadings correlation

Significant correlations are denoted with an asterisk (*).

Reviewer #2, comment 2:

The best strategy to handle the common influence of age, while not throwing the baby out with the water, is challenging. I think the mediation analysis used by the authors is a way of showing the mutual relationship between the variables in a clear way. In the EFA analyses, however, age was regressed out prior to the factor extraction. That may be ok, but it was not entirely clear to me how age was handled in the following analyses.

Authors' response:

We are particularly thankful to this reviewer for the opportunity to clarify this point. It made us realize that the manuscript's introduction failed to make completely clear to the reader that we adopted a two-stage study design with regard to the relationship between age, cortical thickness and methylation.

The first analytical step addressed the question whether global methylation patterns are possible mediators of the effect of age on cortical thinning. To this end, ICA was performed first to describe global methylomic patterns. The second stage of the analysis was initiated once such a mediator (*ICA2*) was identified. Importantly, the mediator exerted a significant effect on cortical thinning also after correction for age. Thus, the second stage consisted of studying the effect of the mediator (*ICA2*) - after correction for age- on additional relevant phenotypes, such as regional cortical thickness and related cognitive traits. Consequently, the exploratory factor analysis (EFA) was performed after systematically regressing out age effects from MRI measurements (ROIs) and methylomic patterns.

Once again, we thank the reviewer for the opportunity to clarify this and added the improved description of our approach on pages 4 and 6 of the revised manuscript.

Reviewer #2, comment 3:

I understand that ICA2 patterns were used, but were the age-adjusted thickness estimates also used, or rather the raw thickness estimates based on the age-adjusted factors? If the latter is the case, maybe this could be followed up by a mediation analysis where age is included, since it is a major source of variation that is now hidden. As I read the rationale for the study, understanding age-related changes in cortical thickness is a major objective, and it would thus be informative if age was not just taken out of the equation. Similar concerns regard the memory analyses.

Authors' response:

This point is closely related to comment 2 of this reviewer (see above) and to the two-stage design of our study: The mediator (*ICA2*) exerted a significant effect on cortical thinning also after correction for age. Thus, the second stage consisted of studying the effect of the mediator - after correction for age- on additional relevant phenotypes, such

as regional cortical thickness and related cognitive traits. Importantly, the correlation between memory performance and chronological age in our healthy young population was not significant ($r = -0.05$, $p = 0.18$). This precluded a mediation analysis between ICA2, memory performance and chronological age.

Reviewer #2, comment 4:

It is mentioned that synaptic pruning during development may account for cortical thinning. It could also be mentioned that myelination of lower cortical layers could cause the cortical mantle to appear thinner on the MR scans. This likely accounts for a substantial part of the observed cortical thinning in development.

Authors' response:

We followed the reviewer's suggestion and added this information on page 14 of the revised manuscript.

Reviewer #2, comment 5:

Could the presentation of which samples that were used for the different analyses be more clearly presented, e.g. in a table, figure or flow chart? If I understand correctly, the age-thickness-methylome associations were found in a sample of you participants only? This should be more clearly stated in the ms, with age-ranges also. One easily gets the impression that this is an older-age study also, and I think this needs to be much more directly addressed, and also touched upon in the discussion, because it will have substantial impact on how the results are conceived.

Authors' response:

Absolutely. As requested by the reviewer, the revised manuscript (suppl. Table 1) contains an improved description of the study populations (including age range). We also address this point on page 13 of the revised manuscript.

Reviewer #2, comment 6:

Minor point:

Is it possible to include some for quantitative information in the abstract also, such as sample size and effect sizes? The Fischl et al paper on whole-brain segmentation would not be the most appropriate as the first FreeSurfer-reference here, since the present paper is focused on cortical thickness and the Fischl paper on volume. The 1999 and

2000-papers from Dale and Fischl should rather be referenced first instead.

Authors' response:

The suggested papers are now mentioned on pages 9, 21 and 23 of the revised manuscript. We also put quantitative info in the abstract.

Reply to Reviewer #3

We thank this reviewer for the helpful remarks. We are pleased to inform this reviewer that his/her comments motivated us to study additional samples and to provide additional results, which corroborated our conclusions.

Reviewer #3, comment 1:

The straight-forward analysis that should be presented is an epigenome-wide association analysis (EWAS) testing for association between the methylation of individual CpGs and cortical thickness. This approach allows the reader to gauge the relationship between the 'raw' data and the outcomes.

Authors' response:

We are thankful for the opportunity to clarify this point, as it made us realize that the rationale and the two-stage approach of our study was not made entirely clear in the introduction.

Given that the methylome is a high-dimensional space, at least as much as the transcriptome is, our first goal was to search for genome-wide derived methylomic *patterns*, not singular data points, that might mediate the effect of age on cortical thinning. We were thus interested in gaining a systems-level, genome-wide view of the methylomic signal. Indeed, recent research has demonstrated the importance of analyzing age-associated DNA methylation patterns, as opposed to single CpG sites, when studying the impact of the methylome on physiological processes changing with age, especially when using blood as a surrogate for brain tissue (Horvath et al. *Genome Biology* 2012, 13:R97). Abundant evidence arising from the study of the transcriptome shows that such patterns can be successfully and robustly identified through the use of such decomposition methods as ICA, resulting in the identification of relevant biological processes (e.g. Biton et al., 2014; Rotival et al., 2011; Wexler et al., 2011; for review on ICA see Kong et al, 2008).

Here, we first addressed the question whether methylation patterns are possible mediators of the effect of age on cortical thinning. To this end, ICA was performed first to describe global methylomic patterns (The reason for choosing ICA over other

decomposition methods is described in detail in our response to comment #2 of this reviewer). Using univariate analytical approaches, such as EWAS, where each individual CpG site is independently tested for association with a trait, are ill-suited for detecting methylomic patterns. Said pattern detection can be achieved by projection methods, which decompose the initial high-dimensional dataset into components representing multidimensional DNA methylation patterns amenable to association testing with age-related traits, and amenable to mediation testing.

The second stage of the analysis was initiated once such a mediator (*ICA2*) was identified. Importantly, the mediator exerted a significant effect on cortical thinning also after correction for age. Thus, the second stage consisted of studying the effect of the mediator (*ICA2*) - after correction for age- on additional relevant phenotypes, such as regional cortical thickness and related cognitive traits. Consequently, the exploratory factor analysis (EFA) was performed after systematically regressing out age effects from MRI measurements (ROIs) and methylomic patterns.

Thus, in contrast to decomposed DNA methylation patterns, univariate approaches treating hundreds of thousands of single CpGs as solitary events are ill-suited for addressing the primary question of our study. Once again, we thank the reviewer for the opportunity to clarify this and added the improved description of our approach on pages 4 and 6 of the revised manuscript.

References related to this response:

Biton et al. Independent component analysis uncovers the landscape of the bladder tumor transcriptome and reveals insights into luminal and basal subtypes. *Cell Rep.* 2014 Nov 20;9(4):1235-45.

Horvath et al. Aging effects on DNA methylation modules in human brain and blood tissue. *Genome Biology* 2012, 13:R97

Kong et al. A review of independent component analysis application to microarray gene expression data. *Biotechniques.* 2008 Nov;45(5):501-20.

Rotival et al. Integrating genome-wide genetic variations and monocyte expression data reveals trans-regulated gene modules in humans. *PLoS Genet.* 2011;7(12):e1002367.

Wexler et al. Genome-wide analysis of a Wnt1-regulated transcriptional network implicates neurodegenerative pathways. *Science Signalling.* 2011 Oct 4;4(193):ra65.

Reviewer #3, comment 2:

The authors instead use independent component analysis (ICA) and test the resulting components, without explaining why they jumped to such a global analysis and why specifically ICA. It seems ICA is an alternative to the more common principle component analysis (PCA) when the requirements for PCA are not met: the data are non-Gaussian and (very) noisy. The authors do not state why they used ICA but show data that indicate that the data are indeed quite noisy which raises doubts about the validity of the data sets (an may preclude an EWAS-type analysis):

(1) ICA decomposition resulted in 126 independent component, 111 (88%) of which were driven by a single individual. This suggests that 111 of the 533 individuals actually represent outliers perhaps due to technical issues and probably should have been omitted from the analysis.

(2) SVA identified 40 surrogate variables which is very large considering the sample size and indicates significant technical variability.

Authors' response:

We thank the reviewer for this comment and for pointing out that the reason for choosing the specific method (i.e. ICA) was not well-described. As mentioned in the response to comment #1, we performed ICA to achieve a low-dimensional decomposition of independent age-associated DNA methylation patterns and to investigate their relationship to cortical thickness, an age-related complex trait. Both PCA and ICA represent such decomposition methods, yet they rely on different properties of the inferred components. The choice of ICA over PCA was driven by theoretical assumptions regarding the generative model of observed DNAm signals: under this model, the observed DNAm profiles are viewed as a mixture of independent biological processes. By construction, statistical independence of the *inferred components* implies their non-gaussianity: each component will be characterized by a restricted number of variables, i.e. CpG sites. PCA is based solely on a maximum variance criterion: the initial dataset is projected in a sub-space of successive orthogonal components, each capturing the maximum remaining data variance, irrespectively of any specific generative model underlying the investigated data. In contrast, ICA relies on statistical independency of the components: the observed signal, e.g. DNAm, is assumed to arise from a set of statistically independent sources, which can be viewed as sets of putative independent biological processes influencing the methylome. By assuming statistical

independence, ICA decomposition will favor non-gaussian distribution of the inferred components, thus representing independent sources, or processes, acting on circumscribed sets of features (in our case, CpG sites). Thus, ICA decomposition represents a more realistic approach to the analysis of biological data. This approach, assuming non-gaussianity of underlying sources generating the observed signal, has been repeatedly shown to outperform variance-based decomposition approaches for the analysis of gene expression data. (Liebermeister W, 2002; Lee SI & Batzoglou S, 2003; Frigyesi et al, 2006; Teschendorff et al, 2007) and has been successfully applied to multiple DNA methylation datasets (Renard E et al, 2014).

Further, this comment motivated us to include an independent sample and test the between-sample overlap of ICA methylation patterns. To this end, we performed ICA analysis (for detailed description, see Methods below attached to this response) of the study population reported in Hannum et al., 2013 (Hannum G. et. al. Genome-wide methylation profiles reveal quantitative views of human aging rate. *Mol Cell*. **49**, 359-67. (2013)), which consists of n=656 blood DNAm profiles of participants spanning a wide age-range (19-101, mean age: 64 years).

In the first step of the analysis we identified 5 ICA patterns that were significantly associated with age (designated HIC1a-e, see Table 1 in this response). In the second step we examined the overlap between *ICA1* and *ICA2* CpGs identified in our sample of healthy young adults and CpGs contributing to each of the Hannum age-associated IC pattern. A significant overlap with *ICA1* was observed for one pattern (OR=90, $p < 1e-60$, see Table 1 in this response). For *ICA2* we observed a significant overlap with three age-associated Hannum patterns (see Table 1 in this response), with said overlap being particularly strong for Hannums' HICd pattern (OR=48.7, $p < 1e-60$). Importantly, the correlation of loadings between CpGs contributing to *ICA2* and CpGs contributing to Hannums' HICd pattern was positive and substantial ($r=0.87$, $p < 1e-60$).

Thus, we observed highly significant between-sample overlap of ICA patterns despite the differences in age structure of these samples, supporting the robustness of the methods presented herein. We added this data on pages 7 and 8 of the revised manuscript.

The reviewer also raised an issue regarding the number of surrogate variables inferred

from SVA (40 variables in our sample), considering this number to be possibly too high and indicative of possible data noisiness. We would like to stress that the number of inferred components is both a function of sample size and dimensionality of the dataset (450000 CpG sites in our case). Thus, it is expected that the number of inferred surrogate variables from the 450K array may largely exceed the number of surrogate variables determined in lower-dimensionality datasets such as gene expression data or the 27K methylation array. Importantly, we followed the same SVA approach in Hannum's dataset (450K array, 656 subjects). As expected, we determined a higher number ($n=98$) of surrogate variables in this sample. We added this data on page 22 of the revised manuscript.

Finally, a question was raised regarding the number of retained independent components. We applied stringent criteria and discarded from further analysis every component whose variance was explained by more than 10% by one single individual. Such patterns may or may not represent singular modes of variation, which renders their interpretation difficult at the population level. Importantly, examination of the RLE and Multidimensional Scaling plots did not reveal outlying patterns for the 111 individuals (Figures 1 and 2 in this response) that would call for an *a priori* exclusion of these subjects from analysis.

In addition, we examined the association of *ICA2* with age and age-adjusted cortical thickness considering the 111 individuals as a group covariate: both associations remained highly significant (age: $p = 4.91e-12$; age-adjusted cortical thickness: $p = 4.8e-5$). We added this data on page 8 of the revised manuscript.

Table 1: Age-associated ICA DNAm patterns in the Hannum et al study.

Hannum's IC	Association between age and IC pattern		# CpGs (a)	CpGs Loadings overlap for ICA1		CpGs Loadings overlap for ICA2	
	r	p		OR (b)	p (c)	OR (b)	p (c)
HICa	0.68	<1e-60	2493	90.9	<1e-60	3.5	1.9e-6
HICb	0.35	<1e-60	3236	0	1	0.12	0.99
HICc	0.26	1.7e-11	2142	0.25	0.98	9.4	2.4-e29
HICd	0.17	1.3e-05	1766	0.60	0.83	48.7	<1e-60
HICe	-0.16	4.3e-5	3210	0.17	0.99	0	1

(a): Number of contributing CpGs for Hannum's age associated IC.

(b): Odds-ratio for a fisher's exact test testing enrichment of ICA1/ICA2 CpGs in Hannum's signature; (c): p-value for fisher's exact test.

Methods related to this response:

We analyzed whole-blood methylomic profiles from n=656 samples published in Hannum et al., 2013. In analogy to our methylomic dataset, multi-mapping or polymorphic probes were excluded from analysis. Raw intensities (methylated and unmethylated signals) were normalized using the lumi package (color-bias adjustment and quantile normalisation). The BMIQ algorithm was finally applied to adjust for the difference between Type I and Type II probes used in the 450K array. Given substantial non-randomness of between-plate distribution of chronological age in this sample, we performed CoMbat adjustment for plate effect. DNA methylation values were subsequently adjusted for sex and 98 surrogate variables inferred from surrogate variable analysis (SVA).

ICA decomposition on the adjusted signals yielded a total of 175 components, among which 19 were retained based on the per-subject 10% variance criterion used in our methylomic dataset. The retained ICA patterns were tested for association with age, under adjustment for estimated cell counts (CD4T, CD8T, NK, Gran, Mono, Bcell). Five patterns were significantly associated with age. In analogy to our study, CpGs contributing to these patterns were chosen so as to exhibit an absolute loading > |4| on the respective pattern.

References related to this response:

Frigyesi, A., Veerla, S., Lindgren, D., Höglund, M. Independent component analysis reveals new and biologically significant structures in micro array data. *BMC Bioinformatics*. 7, 290. (2006).

Hannum G et al. Genome-wide methylation profiles reveal quantitative views of human aging rates. *Mol Cell*. 2013 Jan 24;49(2):359-67

Lee, S.-I., & Batzoglou, S. Application of independent component analysis to microarrays. *Genome Biology*. 4,R76 (2003).

Liebermeister, W., Linear modes of gene expression determined by independent component analysis. *Bioinformatics*. 18, 51-60. (2002).

Renard E, Teschendorff AE, Absil PA. Capturing confounding sources of variation in DNA methylation data by spatiotemporal independent component analysis. in *Proc ESANN 2014*, pp. 195–200.

Teschendorff, A., Journée, M., Absil, A.P., Sepulchre, R., Caldas, C. Elucidating the altered transcriptional programs in breast cancer using independent component analysis. *PLoS Computational Biology*, 3, e161. (2007).

Figure 1 RLE plot of SWAN normalized beta values across samples. Values outside 1.5 interquartile range not shown. The 111 individuals associated with possible singular modes of variation are depicted in blue.

Figure 2 Multidimensional Scaling analysis on SWAN normalized beta values across samples. Five first dimensions are shown. The 111 individuals associated with possible singular modes of variation are depicted in blue.

Reviewer #3, comment 3:

Since the primary research question relates to the association of whole blood DNA methylation profiles with cortical thickness, SVA should be performed using cortical thickness as outcome, not age. Now, the interpretation of associations of ICA components remains unclear: is it driven by age, an age-effect on cell composition etc?

Authors' response:

This comment is tightly linked to comment #1 of this reviewer and to the fact that the two-stage approach of our study was not made entirely clear in the introduction. The objective of the study was to investigate the relationship between cortical thickness and age-associated DNAm patterns. Considering age as an outcome in the SVA allowed adjusting DNAm for unknown technical/biological confounders, while keeping the influence of age on DNAm intact.

Reviewer #3, comment 4:

If the primary research question was to gain insight in the role of age in the decline of cortical thickness (as perhaps suggested in the introduction), a specific analysis of known age-associated CpGs (or scores based on multiple CpGs) in blood would have been in order (as mentioned in introduction).

Authors' response:

To address the reviewer's comment, we first applied Horvath's cross-tissue- and Hannum's whole-blood-based predictor in our sample. Both predictors yielded DNA methylation age values that significantly correlated with actual participants' age (Horvath's predictor: $r=0.70$, $p < 1e-60$; Hannum's predictor: $r=0.71$, $p < 1e-60$). Neither predictor was associated with cortical thickness after adjustment for chronological age (Horvath's: $r= 0.04$, $p= 0.32$; Hannum's: $r=0.01$, $p =0.77$), suggesting that these predictors (like *ICA1* but, importantly, unlike *ICA2*) do not mediate the effect of age on cortical thickness. We added this information on page 7 of the revised manuscript.

Reviewer #3, comment 5:

The 15 ICAs not driven by a single individual are not described. How much of the

variability do they explain? What is their nature: with which (technical, cell count variable etc) variables do they correlate? It is elemental to know which CpGs contributed to the ICAs to contribute to our biological understanding. Are they known to be associated with age or differentially methylated between fine grained cell counts?

Authors' response:

We thank the reviewer for this comment. It was due to brevity that we hadn't included information on all 15 ICAs in the first version of the manuscript. Detailed information is now provided in the supplementary information. In this response to the comment we primarily focus on the most important features of the age-associated ICAs (i.e. ICA1 and ICA2).

Examination of the relationship of *ICA2* with known technical confounders (e.g. plate, sentrix position), or gross cell count measures did not show any relevant association (supplementary Table 12 of the revised manuscript and revised methods section).

Using *in silico*-based sub-cell count predictions, we observed significant associations with *ICA2* (*CD4T* accounting for 7.4%, *CD8T* accounting for 4.0%, and NK accounting for 2.6% of *ICA2* variance, supplementary Table 12 of the revised manuscript). Importantly, after adjusting *ICA2* for these blood sub-cell types, the associations with both chronological age and cortical thickness remained highly significant (age: $r=0.29$, $p=2e-11$; cortical thickness: $r=-0.22$, $p=8.3e-7$).

Gene-set enrichment analysis for CpGs contributing to *ICA2* can be found in supplementary Table 8 of the revised manuscript. Results showed significant enrichment for 76 highly overlapping pathways (FDR<0.05), notably related to immune system development and function.

These results suggest that association between *ICA2* and cortical thickness is not driven by major or subtle changes in blood composition. Nevertheless, *ICA2* harbors primarily CpGs known to influence immune function. We comment on this important feature of *ICA2* on pages 7, 15 and 16 of the revised manuscript.

Age accounted for 8.6% of variability of *ICA2*. We also examined whether *ICA2* harbors CpG markers known to be associated with age. Among the 970 CpGs constituting *ICA2*, one marker (cg18055007) was part of the 353 Horvath predictor markers, and four (cg20822990, cg16054275, cg16867657, cg21139312) were part of the 71 CpGs

included in Hannum's DNAm age model. We added this information on page 7 of the revised manuscript. After adjustment for chronological age, *ICA2* accounted for 3.1% and 1.7% of variance of cortical thickness and spatial *F6* factor score, respectively (see Table 1 in this response).

Regarding *ICA1*, chronological age accounted for 29.7% of variance. This methylomic pattern did not show any significant association with technical covariates nor blood cell counts (measured or estimated) (see supplementary Table 12 of the revised manuscript).

Gene-set enrichment analysis of the 739 CpGs constituting *ICA1* (at FDR<0.05) revealed two gene-sets constituting target genes of Polycomb proteins, in line with previous studies reporting marked age-dependent DNAm changes in target genes of Polycomb proteins (Horvath et al., 2012; McClay et al., 2014) (Supplementary Table 13 of the revised manuscript). We also observed strong overlap of *ICA1* contributing CpGs with published age-associated markers (26 CpGs from Hannum or Horvath's DNAm age predictor, Fisher's exact test enrichment $p < 3e-30$).

The same analyses were conducted for the remaining 13 ICs (supplementary Tables 12 and 13 of the revised manuscript). However, without a significant association of these ICs with the phenotypes under study (i.e. age and cortical thickness), interpretation of the results is limited.

Table 1: Fraction of variance shared between *ICA2* pattern and primary phenotypes.

Variable	Fraction of variance (r^2 in %)
Age	8.6
Cortical thickness	3.1
Thickness Factor F6	1.7

References related to this response:

Horvath, S. et al. (2012). Aging effects on DNA methylation modules in human brain and blood tissue. *Genome Biol.* **13**, R97.

McClay, J. L. et al. A methylome-wide study of aging using massively parallel sequencing of the methyl-CpG-enriched genomic fraction from blood in over 700 subjects. *Hum. Mol. Genet.* **23**, 1175–1185 (2014).

Reviewer #3, comment 6:

The main PCAs in whole blood 450k data usually capture cell counts (beyond the larger classes like the 3 measured here). It is not unlikely that the ICAs also represent cell counts. Hence, if true, the data may suggest an association between the immune system (measured through DNA methylation instead of direct cell counts) and cortical thickness. If this association survives robust control for confounding (i.e. including parameters known to be associated with cortical thickness (and immune system traits) (social economic position, medication use, etc.)). This in fact may be a very interesting observation but requires much more specific insight in the biology of the ICAs (if any).

Authors' response:

We thank the reviewer for this important comment. Examination of the relationship of *ICA2* with known technical confounders (e.g. plate, sentrix position), or gross cell count measures did not show any significant association ($p > 0.01$, Supplementary Table 12 of the revised manuscript).

Using *in silico*-based sub-cell count predictions, we observed significant associations with *ICA2* (*CD4T* accounting for 7.4%, *CD8T* accounting for 4.0%, and NK accounting for 2.3% of *ICA2* variance, Supplementary Table 12 of the revised manuscript). Importantly, after adjusting *ICA2* for these blood sub-cell types, the associations with both chronological age and cortical thickness remained highly significant (age: $r=0.29$, $p=2e-11$; cortical thickness: $r=-0.22$, $p=8.3e-7$).

Gene-set enrichment analysis for CpGs contributing to *ICA2* can be found in supplementary Table 8 of the revised manuscript. Results showed significant enrichment for 76 highly overlapping pathways ($FDR < 0.05$), notably related to immune system development and function.

These results suggest that association between *ICA2* and cortical thickness is not driven by major or subtle changes in blood composition. Nevertheless, *ICA2* harbors primarily CpGs known to influence immune function. We comment on this important feature of *ICA2* on pages 7, 15 and 16 of the revised manuscript.

Reviewer #3, comment 7:

Did the authors validate the link between methylation and genotype data (e.g. were they from the same person)?

Authors' response:

Absolutely. A per-subject crosscheck between phenotypic data, methylation data and genetic data was performed using the reported sex and sex-predictions based on the array data, as well as matching of all SNPs represented on the Illumina 450K array to the corresponding Affymetrix SNP 6.0 genotype calls. This crosscheck allowed an unambiguous assignment of each methylation dataset to the corresponding genetic and phenotypic dataset. We added this information on page 25 of the revised manuscript.

Reviewer #3, comment 8:

Using genetic variation as causal anchor is a strong approach. However, identifying robust genetic variants requires a GWAS and hence many thousands if not 10 thousands of samples (instead of 500). Were there GWASs performed for cortical thickness the authors can rely on? Gene-set enrichment approaches as applied here reduce the enthusiasm for a genetic approach and limits its interpretation.

Authors' response:

We would like to stress that we did not perform a GWAS on cortical thickness, nor was this the aim of the study. After having identified a methylomic pattern (i.e. *ICA2*) mediating the effect of age on cortical thinning, we searched for possible genetic patterns that might explain in part this epigenetic phenotype's variability. To this end we capitalized on the power of gene-set enrichment analysis (GSEA) to detect genetic pathways that are associated with the trait of interest. As shown repeatedly by us (e.g. Heck et al., 2014; Heck et al., 2015) and others (e.g. O'Dushlaine et al., 2015; Holmans et al., 2009), GSEA identifies robustly genetic patterns relevant to cognitive and neuropsychiatric traits.

References related to this response:

Heck et al. Converging genetic and functional brain imaging evidence links neuronal excitability to working memory, psychiatric disease, and brain activity. *Neuron*. 2014 Mar 5;81(5):1203-13.

Heck et al. Genetic Analysis of Association Between Calcium Signaling and Hippocampal Activation, Memory Performance in the Young and Old, and Risk for Sporadic Alzheimer Disease. *JAMA Psychiatry*. 2015 Oct;72(10):1029-36.

Holmans et al. Gene ontology analysis of GWA study data sets provides insights into the biology of bipolar disorder. *Am J Hum Genet*. 2009 Jul;85(1):13-24.

O'Dushlaine et al. Psychiatric genome-wide association study analyses implicate neuronal, immune and histone pathways. *Nature Neurosci*. 2015 Feb;18(2):199-209.

Reviewer #3, comment 9:

SWAN sometimes induces strange DNA methylation values. It will be useful to exclude such artefacts by comparing the raw DNA methylation values to those post-SWAN.

Authors' response:

As expected, SWAN normalization successfully reduced the technical difference between Infinium probe types (Figure 1 to this response). To rule out systematic shift in DNA methylation values induced by SWAN normalization we compared the correlation between summary statistics of CpGs sites before and after normalisation. We observed high correlation for both average ($r > 0.99$) and variance ($r > 0.95$) of DNA methylation values across samples (Figure 2 to this response).

We also observed high average correlation between DNAm values before and after normalisation per-CpG site (average $r = 0.87$), and per-sample (average $r = 0.89$ after mean-centering DNAm values per CpG). We added this information on page 19 of the revised manuscript.

Figure 1: Density plot of DNA methylation values before and after SWAN normalisation averaged across samples.

red curve: type I probes. blue curve: type II probes.

Figure 2: Summary statistics of DNAm values before and after normalisation for each probe type.

Upper panel: Mean beta values across samples for all CpG sites.

Lower panel: Variance of beta values across samples for all CpG sites.

Reviewer #3, comment 10:

How many methylation values were missing and imputed? What were setting of the impute package?

Authors' response:

The average per CpG and per sample missing rates were $< 1e-04$; CpGs with missing rate > 0.05 were excluded from analysis. The impute package was used by taking default settings (k=10 nearest neighbors).

Reviewer #3, comment 11:

DNA methylation data seems to be generated on a sample taken ~1 years after the MRI. It should be discussed how this affected the study.

Authors' response:

DNA methylation profiles were obtained on average 1 year after imaging acquisition. We performed a sensitivity analysis examining the association between *ICA2* and cortical thickness after regressing out the difference (delta age) between age at blood sampling and age at MRI assessment from the methylomic pattern. The association remained significant ($p = 6.4e-5$, $r = -0.18$ after adjustment for age) indicating that delta age did not affect the results of the study. We added this information on page 19 of the revised manuscript.

Reviewer #3, comment 12:

Why were the measured cell counts not included in SVA?

Authors' response:

SVA is recommended as a robust method for cell-type mixture adjustment (McGregor et al, 2016), allowing correction also beyond gross cell composition measures. Importantly, we did not observe significant association of measured blood cell counts with age nor cortical thickness (suppl. Table 11 of the revised manuscript).

References related to this response:

McGregor et al. (2016) An evaluation of methods correcting for cell-type heterogeneity in DNA methylation studies. *Genome Biol.* 17:84.

Reviewer #3 (Remarks to the Author):

I genuinely appreciate the considerable additional work the authors put into the manuscript. I believe this has strengthened the manuscript and improved the clarity of their reasoning. I feel, however, that when adopting non-standard analysis approaches (ICA instead of EWAS) particularly in a novel field like epigenetic epidemiology, the burden of proof that the outcomes are robust is on the authors. I have three major concerns, of which certainly the latter two can be addressed easily.

1. The finding that ICA2 is associated with cortical thickness is based on a single study of 553 individuals. Replication is a key aspect of any genomic association study.

Response to Reviewer # 3-1:

We are particularly thankful to the reviewer for this comment. Motivated by his/her suggestion to address the issue of replication, we reached out to our colleagues at the Max Planck Institute for Psychiatry in Munich. Analyses of the relationship between DNAm and cortical thickness in this sample (termed herein “Munich sample”) were done entirely by the Munich team, i.e. we were not involved in order to ensure full independence of the analytical steps.

The Munich sample comprised N=627 participants who underwent MRI assessment and whole-blood methylomic profiling (423 MDD patients, mean age 47.9 ± 13.8 ; 204 controls, mean age 49.5 ± 13.3). After pre-processing of methylomic data and MRI-QC based exclusion, N=596 subjects remained for association testing of cortical thickness and ICA2 (estimated from whole-blood samples).

The ICA2 pattern was estimated as the linear combination between ICA2 loadings (as inferred from the Swiss DNAm sample) and individual DNAm values of the Munich sample. In this independent sample, we observed a significant positive correlation between ICA2 and chronological age ($r=0.48$, $p < 1e-10$; as a reminder: the corresponding value in the Swiss sample was $r=0.29$) and a negative correlation with global cortical thickness ($r= -0.31$, $p < 1e-10$; as a reminder: the corresponding value in the Swiss sample was $r= -0.24$). After adjustment for chronological age and controlling for potential

confounders (diagnosis, sex, intracranial volume, MR-batch effects, time difference between MRI examination and blood drawing), the association between *ICA2* and cortical thickness remained significant ($r = -0.094$, $p = 0.011$; as a reminder: the corresponding value in the Swiss sample was $r = -0.18$). Importantly, the same analysis in a sub-sample of $N=163$ participants younger than 40 years (thus, within an age range similar to that of the Swiss participants) revealed an almost identical effect size ($r = -0.19$, $p = 0.009$) compared to that observed in the Swiss sample. Thus, we are pleased to report the independent replication of our primary finding and are therefore particularly thankful to this reviewer for motivating us to put this additional effort into our study.

We added these data on page 8 of the revised manuscript.

Methods related to this response:

Munich sample:

Sample description: The Munich sample consisted of patients with first episode and recurrent unipolar depression treated as in-patients at the Max Planck Institute of Psychiatry, Munich, and healthy control subjects ($N=627$ with combined MRI and DNA availability; 423 patients, age 47.9 (SD 13.8) years; control subjects age 49.5 (SD 13.3) years), for the most part overlapping with imaging genetic and MDD association studies reported in collaboration with the ENIGMA consortium (Stein et al., 2012; Schmaal et al., 2016). Other than in the flagship study (Stein et al., 2012), no bipolar patients were included for reasons of clinical homogeneity (Schmaal et al., 2016). MDD diagnoses were based on clinical consensus in addition to M-CIDI or SCAN interviews, depending on the original study protocols. After pre-processing of methylomic data, and MRI-QC based exclusions, combined data of $N=596$ subjects was available for statistical analysis.

Structural imaging: MRI acquisition: High resolution T1-weighted images were acquired at the Neuroimaging Core Unit of the MPIP on a clinical 1.5 Tesla MR scanner (Signa/Signa Excite, General Electric, for sequence details see ^{1,2}). **MRI data processing:** Gross morphological abnormalities such as

tumor or territorial infarction, ventricle asymmetries or arachnoid cysts preventing automated image processing, extensive white matter disease or motion artefacts were exclusion criteria prior to the formation of this combined sample. The surface-based segmentation stream of FreeSurfer (version 5.3, installed on 64-bit Linux workstations) was applied to all T1-weighted images, with substeps as described in the Structural Imaging section. Visual QC of cortical segmentation quality was performed on the basis of standardized protocols (<http://enigma.ini.usc.edu/protocols/imaging-protocols>) and led to exclusion of 12 subjects. As phenotypes of interest, left and right cortical thickness (the average of which is ref. to as cortical thickness [CT]), and intracranial volume derived indirectly from the spatial registration procedure.

Methylomic profiling: DNA was extracted from whole blood using the Genra Puregene Blood Kit (QIAGEN). Quality and quantity of the DNA were assessed by NanoDrop 2000 Spectrophotometer (Thermo Scientific) and Quant-iT Picogreen (Invitrogen). Genomic DNA was bisulfite converted using the Zymo EZ-96 DNA Methylation Kit (Zymo Research) and genome-wide methylome levels were assessed with the Illumina Infinium HumanMethylation 450K BeadChip array. Hybridization and processing was performed according to manufacturer's instructions. Intensity read outs, normalization and estimation and beta values were obtained using the Minfi package (version 1.21.0) in Bioconductor (Aryee et al., 2014). Beta values for the pre-selected 397,947 autosomal probes from the Swiss sample were calculated from SWAN normalized intensities. After pre-processing of methylomic data, and MRI-QC based exclusions, combined data of N=596 subjects was available for statistical analysis.

Association with ICA2 pattern: ICA2 patterns were calculated separately for the whole Munich sample and a subsample of <40-year-old subjects (N=163). DNA methylation values were first adjusted for sex using linear regression. ICA2 patterns were then calculated as linear combination between the scaled residuals and the inverse ICA2 loadings inferred from the Swiss sample. Separate Pearson's correlation analyses were performed between ICA2-scores and biographical age, and ICA2-scores and CT scores. In addition,

partial correlation analyses were performed between ICA2-scores and CT, correcting for age at MRI, difference between age at MRI and age at blood-drawing, sex, intracranial volume and MRI batch effects. All p -values reported in the replication sample are one-sided.

References:

Aryee, M. J. *et al.* Minfi: a flexible and comprehensive Bioconductor package for the analysis of Infinium DNA methylation microarrays. *Bioinformatics* **30**, 1363–1369 (2014).

Schmaal, L. *et al.* Subcortical brain alterations in major depressive disorder: findings from the ENIGMA Major Depressive Disorder working group. *Mol. Psychiatry* **21**, 806–12 (2016).

Stein, J. *et al.* Identification of common variants associated with human hippocampal and intracranial volumes. *Nat. Genet.* **44**, 552-61 (2012).

2. The authors claim that the association of ICA2 with cortical thickness and other phenotypes is not driven by blood cell counts. In general, cell counts are well-known to be main drivers of major components explaining methylomic variance. Accounting for main cell types is not sufficient to exclude this possibility. The fact that ICA2-CpGs are near genes that are primarily involved in processes related to inflammation and leukocyte differentiation is compatible with a cell count effect. As mentioned in my first review, an immune component related to cortical thickness is of interest but we should know its nature. The authors may use the approach adopted in a recent paper on age-related changes in DNA methylation (Slieker et al. *Genome Biol* 2016; 17:191), where the authors took public 450k data from many blood cell subtypes and then assess whether identified CpGs are differentially methylated between cell subtypes. The authors can do this for their 970 ICA2 CpGs.

3. The authors do use a public data set (Hannum et al) to confirm the presence of ICA1 and 2 and their correlation with age. This data-set was, however, on whole blood. They should extend this analysis to public data on purified blood cells like the one published in *Nat Commun* covering monocytes and T cells (Reynolds et al. *Nat Commun*. 2014; 5:1–8). This would further substantiate the independence of blood cell counts.

Response to Reviewer # 3-2 & 3-3:

We thank the reviewer for suggesting to provide further evidence for the independence of the results on blood cell counts and for proposing to look deeper into the putative nature of the relation between immunity and cortical thickness by using the data reported in Slieker et al. Here we followed these suggestions and respond to both comments in common because they are closely related.

In addition to the originally reported (Supplementary Table 11) correction for effectively measured major cell type counts, we performed sensitivity analyses of the detected associations using *in silico* annotation of blood sub-cell types (i.e., CD4T, CD8T, NK, granulocytes, monocytes, and B-cells) as described by Jaffe & Irizarry, 2014. After this adjustment, the associations of ICA2 with both chronological age and cortical thickness remained highly significant ($p=2e-11$ and $p=8.3e-7$, respectively). We also examined the

association between *ICA1* and *ICA2* and chronological age in two publicly available datasets of purified blood cells ($N=1202$ monocyte samples, age range: 44-83, mean age: 60; $N= 214$ CD4+ T-cell samples, age range: 45-79, mean age: 59)(Reynolds et al., 2014). In each dataset, *ICA1* and *ICA2* were estimated as the linear combinations between *ICA1* and *ICA2* loadings, respectively (as inferred from the Swiss DNAm sample), and blood samples' DNAm values, adjusted for main confounders (see Methods related to this response). In both cell-specific datasets, we observed a significant positive correlation between *ICA* patterns and chronological age (monocyte samples: *ICA1*: $r= 0.67$, $p < 2.2e-16$; *ICA2* $r= 0.32$, $p < 2.2e-16$; CD4+ T-cell samples: *ICA1*: $r= 0.70$, $p < 2.2e-16$; *ICA2*: $r= 0.49$; $p = 8.6e-15$), suggesting that the *ICA*-age correlations identified in whole-blood are also detectable in individual cell types. Altogether, these results substantiate the lack of influence of blood cell counts on the reported associations.

The reviewer also proposed to explore further the nature of the immune component related to cortical thickness. Following the reviewer's suggestion, we compared the DNA methylation of the 970 most prominent *ICA2* CpGs to that of blood cell subtypes and their progenitors using public data sets on 19 cell types (see Methods related to this response) (Slieker et al., 2016). We observed consistently highly significant correlations ($p < 1e-60$ for all correlations) between average whole-blood DNAm values of the *ICA2* CpGs and all various cell subtypes examined (Figures 1-3 of this response). The lowest correlation coefficients (albeit still highly significant with $p < 1e-60$) were observed for regulator and memory CD4+ T-cells (Figure 3). Generally, the correlation coefficients might suggest high concordance of the cortical thickness-related blood DNAm patterns with DNAm of B lymphocytes and of the common myeloid progenitor lineage, and relatively less concordance with DNAm of natural killer cells and T lymphocytes. Given the correlative nature of all these analyses, we suggest being humble in our interpretation and report this analysis in the main body of the manuscript by highlighting the cautiousness, which should be applied to such conclusions and stating that no

further inference can be drawn towards the contribution of a specific immune cell type to the reported associations.

Altogether, we are truly thankful to this reviewer for his/her input, which led to significant improvement of the manuscript. We now provide independent replication of the primary finding and show that the results are independent of blood cell composition-related bias. The new results can be found on pages 7, 11 and 12 of the revised manuscript.

References:

Absher DM et al. (2013) Genome-wide DNA methylation analysis of systemic lupus erythematosus reveals persistent hypomethylation of interferon genes and compositional changes to CD4+ T-cell populations. *PLoS Genet* 9(8):e1003678.

Jaffe A., Irizarry RA. (2014) Accounting for cellular heterogeneity is critical in epigenome-wide association studies. *Genome Biol* 15(2):R31.

Jung N et al. (2015) An LSC epigenetic signature is largely mutation independent and implicates the HOXA cluster in AML pathogenesis. *Nat Comm* 7(6):8489.

Reinius LE et al. (2012) Differential DNA methylation in purified human blood cells: implications for cell lineage and studies on disease susceptibility. *PLoS One* 7(7):e41361.

Reynolds LM et al. (2014) Age-related variations in the methylome associated with gene expression in human monocytes and T cells. *Nat Comm* 18(5): 5366.

Slieker RC et al. (2016) Age-related accrual of methylomic variability is linked to fundamental aging mechanisms. *Genome Biol* 17(1): 191.

Vento-Tormo R et al. (2016) IL-4 orchestrates STAT6-mediated DNA demethylation leading to dendritic cell differentiation. *Genome Biol* 17:4.

Figure 1: Average whole-blood DNAm at 970 *ICA2* CpGs versus progenitor cell specific DNAm.

Horizontal axis: average whole-blood DNAm observed in the methylomic Swiss sample (n=533). Vertical axis: average DNAm observed in progenitor cells.

Figure 2: Average whole-blood DNAm at 970 *ICA2* CpGs versus cell subtypes specific DNAm.

Horizontal axis: average whole-blood DNAm observed in the methylomic Swiss sample (n=533). Vertical axis: average DNAm observed in specific cell subtypes.

Figure 3: Average whole-blood DNAm at 970 *ICA2* CpGs versus lymphocytes subtypes specific DNAm.

Horizontal axis: average whole-blood DNAm observed in the methylomic Swiss sample (n=533). Vertical axis: average DNAm observed in specific cell subtypes.

Methods:

Association of ICA patterns with chronological age in cell-specific methylomic profiles: We used publically available methylomic profiles from N=1202 monocytes samples (GSE56046) and N=214 CD4 T-cells samples (GSE56581)(Reynolds et al., 2014) Normalized datasets deposited on GEO repository were considered for analysis. In each dataset a Surrogate Variable Analysis preserving for chronological age was performed. Individual methylomic values were adjusted for the inferred SVs using linear regression. In each dataset, *ICA1* and *ICA2* patterns were estimated as the linear combination between the inverse of genome-wide *ICA1* and *ICA2* loadings (inferred from the Swiss sample) and scaled SV-adjusted DNAm values. This score was subsequently tested for association with chronological age.

Comparison of whole-blood and cell-specific DNAm values: Average DNA methylation values were obtained from four publically available datasets from 19 cell types. Average DNAm values from hematopoietic stem cells and progenitor cells were obtained from GSE63409 (Jung et al. 2015) considering only normal bone marrow samples. Average DNAm from whole-blood, PBMCs, Natural Killer cells, B-lymphocytes, CD4 T-cells, CD8 T-cells, monocytes, neutrophils, eosinophils and granulocytes were obtained from GSE3560 (Reinius, et al., 2012). Average DNAm from specific sub-types of CD4 T-cells (naive, memory and regulatory CD4 T-cells) were obtained from GSE59250 (Absher, 2013) considering control samples only. Average DNAm in dendritic cells and macrophage (in vitro induced) were obtained from GSE75937 (Vento-Tormo, 2016).

Reply to Reviewer #4

We gratefully appreciate the positive feedback and the constructive remarks made by this reviewer.

Reviewer #4, comment 1:

A justification as to why ICA was applied in favour of other more established analytical approaches could be added. In addition, the authors could elaborate on the limitations of ICA in the Discussion section.

Authors' response:

We are thankful for the opportunity to elaborate on the importance of the use of decomposition methods when analyzing methylomic data and also to discuss weaknesses of ICA (please see page 17 of the revised manuscript):

Given that the methylome, as any –omic dataset, is a high-dimensional space, our goal was to search for genome-wide derived methylomic *patterns*, not singular data points, that might mediate the effect of age on cortical thinning. Thus, we were interested in gaining a systems-level, genome-wide view of the methylomic signal. Indeed, recent research has demonstrated the importance of analyzing age-associated DNA methylation *patterns*, as opposed to single CpG sites, when studying the impact of the methylome on physiological processes changing with age, especially when using blood as a surrogate for brain tissue (Horvath et al. Genome Biology 2012, 13:R97). Abundant evidence arising from –omics studies shows that such patterns can be successfully and robustly identified through the use of such decomposition methods as ICA, resulting in the identification of relevant biological processes (e.g. Biton et al., 2014; Rotival et al., 2011; Wexler et al., 2011; for review on ICA see Kong et al, 2008). Standard univariate analytical approaches, such as EWAS, where each individual CpG site is independently tested for association with a trait, are ill-suited for detecting methylomic patterns. Said pattern detection can be achieved by projection methods, which decompose the initial high-dimensional dataset into components representing multidimensional DNA methylation patterns amenable to downstream association testing with age-related traits.

Both PCA and ICA represent such decomposition methods, yet they rely on different properties of the inferred components. The choice of ICA over PCA was driven by theoretical assumptions regarding the generative model of observed DNAm signals: under this model, the observed DNAm profiles are viewed as a mixture of independent biological processes. By construction, statistical independence of the *inferred components* implies their non-gaussianity: each component will be characterized by a restricted number of variables, i.e. CpG sites. PCA is based solely on a maximum variance criterion: the initial dataset is projected in a sub-space of successive orthogonal components, each capturing the maximum remaining data variance, irrespectively of any specific generative model underlying the investigated data. In contrast, ICA relies on statistical independency of the components: the observed signal, e.g. DNAm, is assumed to arise from a set of statistically independent sources, which can be viewed as sets of putative independent biological processes influencing the methylome. By assuming statistical independence, ICA decomposition will favor non-gaussian distribution of the inferred components, thus representing independent sources, or processes, acting on circumscribed sets of features (in our case, CpG sites). Thus, ICA decomposition represents a more realistic approach to the analysis of biological data. This approach, assuming non-gaussianity of underlying sources generating the observed signal, has been repeatedly shown to outperform variance-based decomposition approaches for the analysis of gene expression data. (Liebermeister W, 2002; Lee SI & Batzoglou S, 2003; Frigyesi et al, 2006; Teschendorff et al, 2007) and has been successfully applied to multiple DNA methylation datasets (Renard E et al, 2014).

On the limitations side, decomposition of genome-wide methylomic profiles comes at the cost of specificity of the inferred solution towards the genomic localization of CpG markers. The detection of CpGs contributing to the methylomic signature relies on a fixed threshold on the distribution of the components' loadings. In our case, this approach allowed relating *ICA2* broadly to genes involved in immune system function. However, the specific relationships between the identified marker sets and the phenotypes of interest can be studied only in downstream experiments focusing on single CpG sites.

It is also important to stress that the ICA model relies on the assumption that methylomic signals arise from a fixed set of independent sources. In absence of a priori knowledge

about the source signal, the number of inferred components must be determined empirically, which might impact negatively on generalizability. Indeed, generalizability is a general challenge of systems-level genome-wide approaches (Ritchie et al., 2015). Integration of multiple-layers of molecular traits, such as genotypic data used in this study, is therefore important to address whether the identified patterns represent relevant features of the dataset.

References related to this response:

Biton et al. Independent component analysis uncovers the landscape of the bladder tumor transcriptome and reveals insights into luminal and basal subtypes. *Cell Rep.* 2014 Nov 20;9(4):1235-45.

Frigyesi, A., Veerla, S., Lindgren, D., Höglund, M. Independent component analysis reveals new and biologically significant structures in micro array data. *BMC Bioinformatics.* 7, 290. (2006).

Horvath et al. Aging effects on DNA methylation modules in human brain and blood tissue. *Genome Biology* 2012, 13:R97

Kong et al. A review of independent component analysis application to microarray gene expression data. *Biotechniques.* 2008 Nov;45(5):501-20.

Lee, S.-I., & Batzoglou, S. Application of independent component analysis to microarrays. *Genome Biology.* 4,R76 (2003).

Liebermeister, W., Linear modes of gene expression determined by independent component analysis. *Bioinformatics.* 18, 51-60. (2002).

Renard E, Teschendorff AE, Absil PA. Capturing confounding sources of variation in DNA methylation data by spatiotemporal independent component analysis. in *Proc ESANN 2014*, pp. 195–200.

Ritchie et al. (2015) Methods of integrating data to uncover genotype-phenotype interactions. *Nat Rev Genetics.* 16(2):85-97.

Rotival et al. Integrating genome-wide genetic variations and monocyte expression data reveals trans-regulated gene modules in humans. *PLoS Genet.* 2011;7(12):e1002367.

Teschendorff, A., Journée, M., Absil, A.P., Sepulchre, R., Caldas, C. Elucidating the altered transcriptional programs in breast cancer using independent component analysis. *PLoS Computational Biology*, 3, e161. (2007).

Wexler et al. Genome-wide analysis of a Wnt1-regulated transcriptional network implicates neurodegenerative pathways. *Science Signalling.* 2011 Oct 4;4(193):ra65.

Reviewer #4, comment 2:

It would be important to add a clarification on how significance thresholds were defined, particularly for the association testing between the ICA components and traits of interest. Is there an appropriate equivalent of 'genome-wide significance'?

Authors' response:

We thank the reviewer for this remark and for the opportunity to report the specific thresholds used for multiple testing correction. When using such decomposition methods as ICA, multiple-correction depends on the number of identified components, which is not known a priori. In our case, the genome-wide methylomic dataset was decomposed into 15 components that were amenable to downstream association testing. Hence, traits correlated with these 15 components were subjected to following α level adjustment: $p = 0.05/15 = 0.0033$.

After having identified *ICA2* as the only pattern associated with cortical thickness, we further investigated its relationship with 8 regional thickness factor scores. The α level was thus adjusted for eight tests conducted ($p = 0.05/8 = 0.00625$).

These adjustment schemes are now clearly mentioned in the Methods section (page 28) of the revised manuscript.

Reviewer #4, comment 3:

*Related to (2): Some reported correlations have highly significant p-values ($p < 2.2e-16$), yet the corresponding correlation coefficients seem low ($r = 0.32$). Some associations are dismissed as speculation, despite reporting $p < 1e-60$ as for the case of whole blood vs. monocyte *ICA2* DNAm values. I am not challenging the statistics, but the authors could make a better effort in reporting the statistics more consistently and with more caution throughout the manuscript.*

Authors' response:

This comment made us realize that it was not always clear to the reader that the reported statistical values referred to different sample sizes. For instance, the correlation coefficient $r = 0.32$ was reported for an external methylomic data set comprising $N = 1202$

subjects, thus resulting in lower p-values, as compared to statistics reported for our N=533 methylomic sample. Further, the p-value $< 1e-60$ relates to correlation tests between average CpG-centered DNAm values across tissues.

To improve clarity, we now report sample sizes together with statistical values for any test involving different sample sizes than our methylomic sample.

Reviewer #4, comment 4:

Supplementary Figure 1: I assume the y-axis indicates proportion and not percentage?

Authors' response:

Absolutely. We thank the reviewer for this remark and updated the Supplementary Figure 1 accordingly.

Reviewer #4, comment 5:

Introduction: "[...] a single nucleotide polymorphism (SNP)-based, genome-wide study of methylome patterns' genetic variability". The sentence structure could be improved

Authors' response:

The sentence now reads: "Significant findings were subjected to further analyses, including functional annotation of CpGs contributing to the observed methylation patterns, testing for pattern association with region-specific cortical thickness and cognitive performance, and a genome-wide investigation of common genetic variations (single nucleotide polymorphisms, SNPs) that contribute to the variability of the methylomic patterns."

Reply to Reviewer #5

We highly appreciate this reviewer's constructive remarks and important suggestions, which helped corroborating the conclusions of our statistical analyses.

Reviewer #5, comment 1:

Two out of 15 ICA methylomic patterns (termed ICA1 and ICA2) were significantly correlated with age but only ICA2 predicted cortical thickness after controlling for the linear effects of age.

Supplemental table 2 suggests that ICA1 captures the linear affects of age. My question is whether ICA2 captures non-linear components of the effects of age. For this purpose, I propose an analyses where cortical thickness is first predicted from a model containing a 5th degree polynomial of age (i.e., $age+age^2+\dots+age^5$). If ICA2 is added to this model, does it significantly increase the overall R^2 ?

I believe, this answer to this question is important because if this test is not significant, the relation between methylation and cortical thickness could be spurious and caused by age affected both cortical thickness and methylation profiles in blood.

Authors' response:

We are particularly thankful for this important remark and for the reviewer's proposal to investigate the impact of non-linear effects of age on the reported associations.

As suggested by the reviewer, we performed an F-test analysis to compare the fit of a model predicting cortical thickness from a 5th degree polynomial of age ($age+age^2+\dots+age^5$) to the fit of the same model augmented by *ICA2*. We observed a highly significant increase in adjusted R^2 with the addition of *ICA2* to the model ($F(1,507)= 15.6, p =8.8e-05$).

Thus, these results indicate that the association between *ICA2* and cortical thickness is not driven by non-linear effects of age. These results have been added on page 7 of the revised manuscript.

Methods related to this response:

An F-test was performed to compare a full model between cortical thickness as dependent variable and a 5th degree polynomial for age and *ICA2* as independent variables to a restricted model without *ICA2*.

Reviewer #5, comment 2:

The authors claim that the epigenetic signature of age on cortical thickness. ($p < 0.001$). This seems too strong given the non-experimental nature of the data and possibility of alternative explanations. A mediator model predicts that the age-cortical thickness correlation equals the product of the age-epigenetic signature correlation times the epigenetic signature-cortical thickness correlation. This model does not seem to hold making the authors to conclude that the epigenetic signature only partly mediated the affects of age. However, I am unsure about this statement, as this is just a decomposition of a correlation and alternative explanations for such data that do not assume mediation. (e.g., there could be 3rd variable affecting both cortical thickness and methylation).

Authors' response:

We thank the reviewer for this comment, which made us realize the importance of pointing out to the readers that the mediation effect was a partial one, and that the observed low p value ($p < 0.001$) is a result of the large sample size.

The mediation model was built on the rationale that dynamic age-related molecular processes, such as epigenetic changes, might underlie the well-established age-related decrease in cortical thickness. The mediation analysis suggests that *ICA2* significantly mediates the effect of age on cortical thickness, albeit, as rightly pointed out by the reviewer, only partially. We fully acknowledge that, given the associative nature of the data, we cannot exclude the possibility that the correlation observed between *ICA2* and thickness might also be partially driven by an additional non-modeled variable. We modified the discussion (page 17 of the revised paper) accordingly and also refer now to the partial nature of the mediation throughout the manuscript.

Reviewer #5, comment 3:

I wonder if other phenotypic information is available to further study ICA2. Examples include information on physical exercise, smoking, diet, health indicators etc. Currently, the interpretation of ICA2 relies heavily on bioinformatics and blood methylation studies of implicate immune system genes, making this result a bit generic.

Authors' response:

We followed the reviewer's suggestion and examined which available variables (i.e. body

mass index (BMI), smoking, alcohol consumption, frequency of cannabis use) were significantly associated with *ICA2* in addition to age. Smoking frequency was also significantly associated with *ICA2* ($r=0.17$, $p=1e-04$) but not with cortical thickness ($r=-0.072$, $p=0.11$). After adjusting *ICA2* for both age and smoking frequency, its association with cortical thickness remained nearly unchanged ($r= -0.17$, $p= 0.00017$). We also examined whether, and to what extent, *ICA2* CpGs ($n=970$) overlapped with those reported as being differentially methylated ($n=2037$) in smokers (Tsaprouni et al., 2014; Besingi and Johansson, 2014; Su et al., 2016; Bauer et al., 2016). This was the case for a small fraction (3%) of the *ICA2* CpGs.

No significant correlations were detected between *ICA2* and alcohol consumption ($p=0.97$), cannabis use ($p=0.1$), or BMI ($p=0.25$).

We added this information on page 7 of the revised manuscript.

Methods related to this response:

Self-reported smoking frequency was measured on a 4-point Likert scale (0= never, 1= occasionally, 2=1-5cigarettes/day, 3=6-20 cigarettes/day, 4= 20 or more cigarettes/day). Self-reported alcohol consumption and cannabis use frequencies were measured on a 3-point Likert scale (0=never, 1=occasionally, 2=daily). Association testing for each indicator was performed using linear regression.

References related to this response:

Bauer M, Fink B, Thürmann L, Eszlinger M, Herberth G, Lehmann I. Tobacco smoking differently influences cell types of the innate and adaptive immune system-indications from CpG site methylation. *Clin Epigenetics*. 2016 Aug 3;7:83. doi: 10.1186/s13148-016-0249-7.

Besingi W, Johansson A. Smoke-related DNA methylation changes in the etiology of human disease. *Hum Mol Genet*. 2014 May 1;23(9):2290-7. doi: 10.1093/hmg/ddt621.

Su D, et al. (2016) Distinct Epigenetic Effects of Tobacco Smoking in Whole Blood and among Leukocyte Subtypes. *PLoS ONE* 11(12): e0166486. doi:10.1371/journal.pone.0166486

Tsaprouni LG, Yang TP, Bell J, Dick KJ, Kanoni S, Nisbet J, Viñuela A, Grundberg E, Nelson CP, Meduri E, et al. Cigarette smoking reduces DNA methylation levels at multiple genomic loci but the effect is partially

reversible upon cessation. Epigenetics. 2014 Oct; 9(10):1382-96.

Reviewer #5, comment 4:

The authors state that DNA methylation age values were significantly correlated with actual participants' age. As DNA methylation age is essentially the deviation from methylation predicted age and chronological age, this should not be the case. It makes me wonder whether this is caused by possible non-linear effects of age remaining in the DNA methylation age values.

Authors' response:

We are thankful for this comment because we realized that the meaning of the term 'DNA methylation age' was not made completely clear to the reader.

By using the term 'DNA methylation age', we refer to the direct outcome of publicly available predictors for chronological age based on methylomic markers. These predictors have been shown to yield estimates for age that are highly correlated with actual participants' age (Hannum et al., 2013; Horvath et al., 2012; Marioni et al., 2015), as also observed in our methylomic sample. We clarify this now in the manuscript (p. 8).

References related to this response:

Hannum G. et al. (2013) Genome-wide methylation profiles reveal quantitative views of human aging rates. Mol Cell 49, 359-67.

Horvath S. et al. (2012) Aging effects on DNA methylation modules in human brain and blood tissue. Genome Biol 13, R97.

Marioni R.E. et al. (2015) DNA methylation age of blood predicts all-cause mortality in later life. Genome Biol 16, 25.

Reviewer #5, comment 5:

A multilocus genetic score reflecting genetic variability of this signature was associated with memory performance ($p=0.0003$) in 3346 young and elderly healthy adults. Was this multilocus score also correlated with cortical thickness?

Authors' response:

Following the reviewer's suggestion, we examined the association between the polygenic score and cortical thickness in the methylomic sample (N=514). No significant correlation was observed with cortical thickness ($r = -0.06$, $p = 0.08$). We added this information on page 16 of the revised manuscript.

Reviewer #5, comment 6:

Previous studies have linked immune system to cortical thickness. I wonder if the authors could speculate a little about how this methylation profile might be useful (if not simply a non-linear effect of age) could further advance the study of (age-related) cognitive decline?

Authors' response:

As suggested by the reviewer, we now discuss the potential use of peripheral methylation profiles in the study of age-related cognitive decline (last paragraph of the revised discussion section): "For example, peripheral markers of systemic inflammation are associated with reduced grey matter volume, both in midlife adults (Marsland et al., 2015) and in the elderly (Satizabal et al., 2012). Moreover, such grey matter reduction seems to mediate the negative effects of peripheral inflammation on age-related cognitive decline (Marsland et al., 2015). It will be interesting to investigate whether the peripheral DNAm profiles identified herein might be used to differentiate between physiological and pathological age-related cognitive decline and cortical thinning."

References related to this response:

Marsland, A. L. et al. Brain morphology links systemic inflammation to cognitive function in midlife adults. *Brain. Behav. Immun.* 48, 195–204 (2015)

Satizabal, C.L., Zhu, Y.C., Mazoyer, B., Dufouil, C., Tzourio, C., 2012. Circulating IL-6 and CRP are associated with MRI findings in the elderly: the 3C-Dijon Study. *Neurology* 78, 720–727.

Reply to Reviewer #5

We gratefully appreciate the positive feedback and the constructive remarks made by this reviewer.

Reviewer #5, comment:

I still think the authors could have provided a more extensive discussion of the possible advantages of the methylation signature over other immune system markers. For example, traditional immune system markers may reflect the present state, it is very well possible that the methylation preserved a record of past immune response. Further, it may provide a more powerful marker than SNPs due to higher correlations with cortical thickness. I guess such a discussion would serve to point out the unique value of their findings and contribution to the existing literature.

Authors' response:

We added following text in the discussion:

“It will be interesting to investigate whether direct measurement of the immune factors implicated herein along with traditional blood markers of the immune system will provide additional information with regard to the relation between these immune factors and cortical thickness. We speculate that this might not be the case, given the substantial volatility of such direct measurements, which mostly reflect acute state of the immune system, whereas methylation profiles reflect, at least partially, a record of past immune regulation. Nevertheless, further experimental work is warranted to test this hypothesis.”